# SABRE-FL: Selective and Accurate Backdoor Rejection for Federated Prompt Learning

**Momin Ahmad Khan, Yasra Chandio & Fatima Muhammad Anwar**
University of Massachusetts Amherst
{makhan,ychandio,fanwar}@umass.edu

## Abstract

Federated Prompt Learning has emerged as a communication-efficient and privacy-preserving paradigm for adapting large vision-language models like CLIP across decentralized clients. However, the security implications of this setup remain underexplored. In this work, we present the first study of backdoor attacks in Federated Prompt Learning. We show that when malicious clients inject visually imperceptible, learnable noise triggers into input images, the global prompt learner becomes vulnerable to targeted misclassification while still maintaining high accuracy on clean inputs. Motivated by this vulnerability, we propose **SABRE-FL**[1], a lightweight, modular defense that filters poisoned prompt updates using an embedding-space anomaly detector trained offline on out-of-distribution data. SABRE-FL requires no access to raw client data or labels and generalizes across diverse datasets. We show, both theoretically and empirically, that malicious clients can be reliably identified and filtered using an embedding-based detector. Across five diverse datasets and four baseline defenses, SABRE-FL outperforms all baselines by significantly reducing backdoor accuracy while preserving clean accuracy, demonstrating strong empirical performance and underscoring the need for robust prompt learning in future federated systems.

## 1 Introduction

Federated Learning (FL) McMahan et al. (2017) enables decentralized model training across multiple users while keeping data local, thereby preserving privacy and reducing centralized risks. In FL, clients independently train models on local data and share only model updates with a server, which aggregates them into a global model. Due to its privacy-preserving nature, FL has been adopted in settings like Google's Gboard gbo (2017) for next-word prediction, Apple's Siri tec for automatic speech recognition, and WeBank for credit risk prediction webankcredit. Recent advances have extended FL to support more expressive models, such as vision-language models, by integrating prompt-based learning Zhou et al. (2022b); Guo et al. (2023b); Deng et al. (2024).

Prompt learning is a recent paradigm that adapts large pre-trained models such as OpenAI's CLIP (Contrastive Language-Image Pretraining) Radford et al. (2021) to downstream tasks by optimizing lightweight, learnable input prompts instead of finetuning the full model. Originally developed in centralized settings, prompt learning has shown impressive few-shot generalization, task transferability, and reduced compute cost, particularly with vision-language models Zhou et al. (2022b;a). Motivated by these advantages, recent works have introduced prompt learning into FL Guo et al. (2023b); Weng et al. (2024), giving rise to federated prompt learning (FPL). In FPL, clients independently optimize prompt vectors while keeping the model backbone frozen, and share only these prompts with the server. This design greatly reduces communication and memory overhead and enables efficient cross-client adaptation in multimodal and heterogeneous environments.

Despite its appeal, FL is not inherently secure Kairouz et al. (2019). In practice, some clients may behave maliciously, either by corrupting their local training data or manipulating model updates, to influence the behavior of the global model Fang et al. (2020); Shejwalkar et al. (2022); Wang et al. (2020); Shejwalkar & Houmansadr (2021); Baruch et al. (2019); Mhamdi et al. (2018); Xie

---

[1]Github: https://github.com/momin-ahmad-khan/ICLR-SABREFL

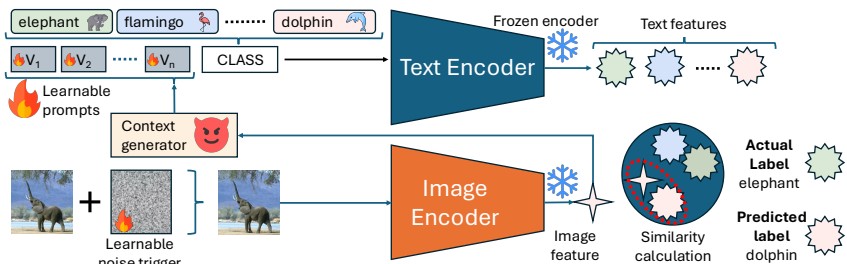

Figure 1: Backdoor attack on prompt-learning-based multimodal models. A learnable and imperceptible noise trigger is added to the image that results in a poisoned image embedding, which is then used to generate the learnable prompts. This addition of noise causes the image features to deviate from their respective text features in the embedding space, thereby causing misclassification.

et al. (2019b); Bhagoji et al. (2019); Bagdasaryan et al. (2020a). A particularly insidious example is the *backdoor attack* Nguyen et al. (2023); Zhang et al. (2023); Bagdasaryan et al. (2020b); Wang et al. (2020), in which an adversary injects carefully crafted inputs (called triggers) into local training data (Figure 1). These triggers cause the global model to misclassify specific test-time inputs while preserving high accuracy on benign samples. Prior work on backdoor attacks in FL has largely focused on traditional classification tasks in unimodal settings Nguyen et al. (2023); Zhang et al. (2023), leaving the security properties of multimodal and prompt-based FL systems underexplored.

While prompt learning in FL is gaining momentum, its *security properties remain largely unexamined*. This raises a key question: **how vulnerable is Federated Prompt Learning to backdoor attacks**? In this work, we show that prompt learners in FL are highly susceptible to backdoors, even when model updates are limited to prompt vectors. We introduce a backdoor attack that inserts a learnable, visually imperceptible trigger into a subset of clients' training data. The attack draws inspiration from BadClip Bai et al. (2024) who design a trigger-based backdoor attack for prompt-learning in the centralized setting. In our attack each malicious client has its own malicious trigger that pushes the prompt embeddings toward a target label in CLIP's semantic space, leading to high-confidence misclassification at inference. The attack remains stealthy and retains high clean accuracy across clients, matching performance observed in centralized prompt tuning. This demonstrates that Federated Prompt Learning is *vulnerable* to trigger-based backdoor attacks even when a few clients act maliciously. To the best of our knowledge, we are the first to study backdoor attacks in this setting, i.e., trigger-based attacks in multimodal federated prompt learning.

Motivated by this, we design **SABRE-FL** (*Selective and Accurate Backdoor Rejection*), a lightweight server-side defense tailored for prompt-based FL. Our key insight is that backdoored prompt vectors yield representations that deviate from the distribution of clean data in CLIP's embedding space. SABRE-FL trains a detector offline, on an out-of-distribution dataset, to recognize these deviations. Importantly, the detector does not require access to client data, labels, or downstream tasks. By leveraging this separation in representation space, SABRE-FL identifies and filters poisoned updates with high precision, maintaining clean model performance while eliminating backdoor impact.

**Contributions:** In our work, we address the critical issue of backdoor attacks in federated prompt learning. In doing so, we make the following key contributions:

- **We introduce the first backdoor attack** specifically targeting prompt learning in FL (§3). The attack injects a visually imperceptible, learnable noise trigger that is optimized to shift prompt representations toward a target class semantically. The attack achieves high backdoor success while preserving clean accuracy, and remains effective even when only a small fraction of clients are compromised, revealing a vulnerability in prompt-based FL systems.
- **Designing SABRE-FL:** We propose SABRE-FL (§4), a lightweight, generalizable defense framework that detects poisoned prompt updates at the server using a classifier trained on out-of-distribution embeddings. We formalize its representation-space decision boundary and provide theoretical conditions for generalization.
- **Comprehensive evaluation and analysis** across five datasets and four defenses; Trimmed Mean, Median, Norm Bounding, and FLAME (§5.1), shows that SABRE-FL consistently outperforms existing methods by achieving lowest backdoor accuracy while maintaining clean accuracy. t-SNE plots and ablations (§5.3) confirm its generalization and effectiveness under diverse FL and prompt learning configurations.

## 2 BACKGROUND AND RELATED WORK

### 2.1 FEDERATED LEARNING (FL)

In FL Kairouz et al. (2019); McMahan et al. (2017), a central entity, known as the *server*, aims to train a *global model*, $\theta^g$, using private data distributed across multiple clients, without directly accessing their data. In each communication round, the server selects $n$ out of $N$ available clients and sends them the current global model $\theta_g^t$, where $t$ denotes the round index. Each selected client $k$ computes an update $\nabla_k^t$ using its local dataset $D_k$, and returns it to the server, which aggregates all updates using a predefined *aggregation rule*, such as FedAvg McMahan et al. (2017).

In *FedAvg*, a client $k$ *fine-tunes* $\theta_g^t$ on their local data using stochastic gradient descent (SGD) for a fixed number of local epochs $E$, resulting in an updated local model $\theta_k^t$. The client then computes their update as the difference $\nabla_k^t = \theta_k^t - \theta_g^t$ and shares $\nabla_k^t$ with the server. Next, the server computes an aggregate of client updates, $f_{\mathsf{agg}}$ using mean, i.e., $\nabla_{\mathsf{agg}}^t = f_{\mathsf{mean}}(\nabla_{\{k \in [n]\}}^t)$ and updates the global model of the $(t+1)^{th}$ round using SGD and server learning $\eta$ as: $\theta_g^{t+1} \leftarrow \theta_g^t + \eta \nabla_{\mathsf{agg}}^t$.

### 2.2 PROMPT LEARNING WITH VISION-LANGUAGE MODELS

**Vision-Language Models:** Large vision-language models (VLMs), such as CLIP Radford et al. (2021), have demonstrated remarkable generalization across diverse downstream tasks. By aligning images and text in a shared semantic space, these models enable strong zero-shot and few-shot performance without task-specific supervision. However, their size, often exceeding hundreds of millions of parameters, makes traditional fine-tuning computationally expensive and bandwidth-intensive, particularly in distributed or resource-constrained environments.

**Prompt Learning Zhou et al. (2022b):** Prompt learning adapts large pre-trained models to downstream tasks by introducing a set of *learnable prompt vectors* that are prepended to the model input. During training, only these prompts are updated, allowing efficient task adaptation while keeping the backbone frozen. This reduces the number of trainable parameters and computational cost, making the approach particularly attractive for few-shot and resource-constrained settings. Prompt learning has been shown to be effective across multiple modalities Khattak et al. (2022); Zhou et al. (2022a;b). In CLIP-based architectures, this involves optimizing a set of context vectors $\boldsymbol{V} = [\boldsymbol{v}_1, \boldsymbol{v}_2, \ldots, \boldsymbol{v}_N]^\top \in \mathbb{R}^{N \times e}$, where each $\boldsymbol{v}_i$ is a learnable token embedding and $e$ is the embedding dimension. Given an input image $\boldsymbol{x}$ and a class name embedding $\boldsymbol{c}_i$, the image encoder $f(\boldsymbol{x})$ and the text encoder $g(\{\boldsymbol{V}, \boldsymbol{c}_i\})$ produce modality-aligned representations. The prediction probability is computed using cosine similarity:

$$p(y = i \mid \boldsymbol{x}) = \frac{\exp(\mathrm{sim}(f(\boldsymbol{x}), g(\{\boldsymbol{V}, \boldsymbol{c}_i\}))/\tau)}{\sum_{j=1}^K \exp(\mathrm{sim}(f(\boldsymbol{x}), g(\{\boldsymbol{V}, \boldsymbol{c}_j\}))/\tau)}, \tag{1}$$

where $\tau$ is a temperature parameter and $\mathrm{sim}(\cdot, \cdot)$ denotes cosine similarity.

**Prompt Learning in FL:** The benefits of prompt learning mentioned above have motivated its integration into the federated setting Guo et al. (2023b;a); Zhao et al. (2023). In *federated prompt learning*, each client optimizes a local prompt vector while keeping the foundation model, e.g., CLIP, frozen, and transmits only the prompt to the server for aggregation. This substantially reduces memory usage and communication cost, making it feasible to deploy foundation models like CLIP in privacy-preserving, bandwidth-limited environments. Such systems have demonstrated strong downstream performance across vision and multimodal tasks while maintaining FL's privacy and scalability benefits.

Despite these advantages, the security implications of prompt learning in FL remain largely unexplored. In particular, it is unclear whether prompt learners, given their limited parameter space and semantic alignment with frozen backbones, are susceptible to adversarial manipulation, such as backdoor attacks. This presents a critical and underexplored vulnerability in the growing area of federated foundation model adaptation.

### 2.3 BACKDOOR ATTACKS

Backdoor attacks Bai et al. (2021; 2022; 2023a); Gao et al. (2023); Li et al. (2021); Turner et al. (2019); Xu et al. (2024); Ya et al. (2024) are a class of training-time data poisoning techniques

Figure 2: An overview of the attack in an FL setting. A malicious client embeds a learnable noise trigger into images. The context generator helps optimize the prompts according to the image features.

wherein an adversary injects carefully crafted samples into the training set with the goal of inducing targeted misbehavior at test time Bagdasaryan et al. (2020a); Hanif et al. (2024). These poisoned samples contain an imperceptible or benign-looking *trigger*, such as a small patch in the input, and are assigned a target label of the attacker's choosing. Once trained on such data, the compromised model behaves normally on clean inputs but misclassifies any input containing the trigger to the target class. This selective misbehavior makes backdoor attacks particularly insidious, as they are challenging to detect using standard validation procedures. Backdoor attacks have been studied across multiple modalities including vision Gu et al. (2017), language Kurita et al. (2020), and multimodal models Liang et al. (2024); Bai et al. (2024) and have proven effective even in privacy-preserving settings like FL, where model updates rather than raw data are shared.

**Backdoor Attacks in Federated Learning:** Backdoor attacks pose a serious security threat to FL, allowing adversaries to embed malicious behavior into the global model by manipulating a small number of clients during training Bagdasaryan et al. (2020b); Wang et al. (2020); Shejwalkar & Houmansadr (2021). These attacks typically preserve high accuracy on clean inputs while causing targeted misclassification on inputs containing an attacker-defined trigger. Early approaches rely on fixed triggers Sun et al. (2019); Xie et al. (2019a); Bhagoji et al. (2019), while more recent methods optimize trigger patterns to maximize attack success and stealthiness Nguyen et al. (2023); Zhang et al. (2023). For example, A3FL Zhang et al. (2023) predicts the movement of the global model updates and improves attack durability by ensuring the backdoor persists across global aggregation rounds. Similarly, IBA Nguyen et al. (2023) jointly optimizes a visually stealthy trigger and selectively poisons models' parameters that are less likely to be updated by the main task's learning process, achieving a durable and stealthy backdoor effect.

## 3 BACKDOOR ATTACKS ON PROMPT LEARNING IN FL

### 3.1 OVERVIEW

While backdoor attacks have been extensively studied in traditional unimodal FL settings, their feasibility in multimodal federated prompt learning remains underexplored. Unlike traditional full-model FL, it exposes a narrower attack surface, limited to prompt vectors, raising new questions about strength, persistence, and stealth of such attacks. These differences motivate our central hypothesis.

**Hypothesis.** We hypothesize that backdoor attacks capable of degrading centralized prompt learning can similarly succeed in federated prompt learning. Despite the distributed setup and aggregation dynamics, prompt-based FL remains vulnerable to targeted manipulation, allowing adversaries to induce misclassifications on trigger inputs while preserving overall model utility on clean data.

**Positioning our work relative to existing literature.** Several recent works have demonstrated the vulnerability of contrastive vision-language models like CLIP to backdoor attacks in centralized settings. Notably, BadCLIP Bai et al. (2024) introduces a powerful trigger-aware attack that jointly manipulates both the image and text encoders using prompt-conditioned triggers. A similar variant Liang et al. (2024) improves stealth and robustness using dual-embedding alignment. Other works such as BadEncoder Jia et al. (2022) and contrastive poisoning attacks Carlini & Terzis (2021) inject backdoors directly into frozen image encoders or pretraining datasets. While effective, these attacks are designed for centralized or pretraining regimes. In this paper, we focus exclusively on the BadCLIP attack due to its compatibility with prompt tuning and its relevance to the downstream FL scenario explored in our work.

**Theoretical Motivation.** In CLIP-based prompt learning Zhou et al. (2022b;a), classification is based on the cosine similarity between an image embedding $f(\boldsymbol{x})$ and a prompt-conditioned text embedding $g(\{\boldsymbol{V}, \boldsymbol{c}_i\})$ for class $i$. To induce targeted misclassification toward a specific class $t$, it suffices to craft an input $\boldsymbol{x}^\star$ such that:

$$\text{sim}(f(\boldsymbol{x}^\star), g(\{\boldsymbol{V}, \boldsymbol{c}_t\})) > \text{sim}(f(\boldsymbol{x}^\star), g(\{\boldsymbol{V}, \boldsymbol{c}_y\})), \quad \forall y \neq t \tag{2}$$

This condition ensures that the model classifies $\boldsymbol{x}^\star$ as belonging to the target class $t$. In practice, our attack injects a visually imperceptible trigger, as shown in Figure 1, into local training data and optimizes it to shift image embeddings toward $g(\{\boldsymbol{V}, \boldsymbol{c}_t\})$, effectively planting a backdoor in the global prompt learner. While this idea is inspired by prior work on backdoor optimization Bai et al. (2024); Zhang et al. (2023); Nguyen et al. (2023), adapting it to the prompt-only FL setting introduces new challenges: the global model is now updated solely via lightweight prompt vectors, and the image encoder remains frozen. This means the backdoor signal must propagate indirectly through prompt aggregation, requiring the trigger to consistently bias prompt updates without direct influence over model weights, making the optimization problem both weaker in signal and more sensitive to noise. The attack is visually explained in Figure 1.

**Evaluation Metrics.** We report two metrics: *Clean Accuracy (CA)* and *Backdoor Accuracy (BA)*. Let $\mathcal{D}_{\text{clean}} = \{(x_i, y_i)\}$ denote the clean test set and $\mathcal{D}_{\text{bd}} = \{(x_i^\star, y_t)\}$ the backdoored test set, where $x_i^\star = x_i \oplus t$ is the triggered input for target label $y_t$. Clean Accuracy, the percentage of clean inputs predicted correctly, is defined as $\text{CA} = \frac{1}{|\mathcal{D}_{\text{clean}}|} \sum \mathbb{K}[\hat{y}(x_i) = y_i]$, while Backdoor Accuracy, the percentage of backdoored inputs predicted as the target label, is $\text{BA} = \frac{1}{|\mathcal{D}_{\text{bd}}|} \sum \mathbb{K}[\hat{y}(x_i^\star) = y_t]$.

## 3.2 THREAT MODEL

**Objective.** The adversary's goal is to perform a targeted backdoor attack in a federated prompt learning setup. By injecting a learnable, visually imperceptible trigger into a subset of training inputs at compromised clients and relabeling them to a fixed target class, the attacker aims to corrupt the global prompt learner. At inference time, inputs stamped with the trigger are misclassified as the attacker's chosen class, while clean inputs remain unaffected, thus maintaining high clean accuracy.

**Capabilities.** We assume a standard FL setup with $N$ clients and a central server aggregating client prompt updates. The adversary controls a fraction $m/N$ of clients, set to 25% by default, consistent with prior works Cao et al. (2023; 2021). The attacker can:

- Modify a subset of local training data by adding a learnable backdoor trigger to inputs.
- Relabel triggered samples to the desired target class, known in literature as *dirty-label* attack Shejwalkar et al. (2023); Gu et al. (2019); Chen et al. (2017); Sarkar et al. (2021); Zeng et al. (2021); Nguyen & Tran (2020); Li et al. (2021).
- Optimize the trigger jointly with the prompt learner at each malicious client to maximize its effect on the global prompt vector.

**Knowledge.** Since the attacker controls client devices, it naturally has access to the full prompt learning setup, including model architecture, frozen CLIP backbone, and training procedure. This is a standard assumption in federated backdoor attack literature Bagdasaryan et al. (2020b); Shejwalkar et al. (2022), and reflects realistic adversaries in open-source or distributed deployments where models like CLIP are publicly available Radford et al. (2021).

## 3.3 DESIGN OF THE BACKDOOR ATTACK IN AN FL SETTING

We illustrate the overall system of the backdoor attack in Figure 2. At the beginning of each communication round, the server distributes (step 1) the current global prompt learner to all participating clients. Unlike traditional FL systems that transmit full model parameters, prompt-based FL transmits only the learnable prompt vectors, significantly reducing communication overhead. The clients keep their model backbones, the image encoder $f_{\text{img}}$ and the text encoder $f_{\text{text}}$, frozen. During local training (step 2), each client fine-tunes the received prompt vectors on its private data. Malicious clients, however, inject a learnable additive noise trigger into a subset of their training images and assign these poisoned samples to an attacker-specified target label, $y_{\text{target}}$. The objective of malicious clients is to optimize their prompt learners such that the presence of the trigger at inference time reliably

causes misclassification, without noticeably affecting clean accuracy. After completing local updates, clients send their locally adapted prompt vectors back to the server (step 3). The server aggregates (step 4) these updates to form the new global prompt learner, which is then redistributed to all clients. This process repeats over multiple rounds until convergence.

**Attack Formalization:** Let $(x, y)$ be a clean image and label pair, with $x \in \mathcal{X}$ and $y \in \mathcal{Y}$. Let $f_{\text{img}} : \mathcal{X} \rightarrow \mathbb{R}^d$ be the image encoder and $f_{\text{text}} : \mathcal{Y} \rightarrow \mathbb{R}^d$ be the text encoder from a frozen CLIP model. Prediction is defined as:

$$\hat{y} = \arg \max_{c \in \mathcal{Y}} \cos(f_{\text{img}}(x), f_{\text{text}}(c)) \tag{3}$$

The attacker injects a learnable trigger $t \in \mathcal{X}$ such that $x^\star = x \oplus t$ is indistinguishable from $x$ in pixel space, but shifts its embedding in CLIP space.

**Goal:**

$$\cos(f_{\text{img}}(x^\star), f_{\text{text}}(y_{\text{target}})) > \cos(f_{\text{img}}(x^\star), f_{\text{text}}(y)) \tag{4}$$

This causes the model to predict $y_{\text{target}}$ instead of the true label $y$. The trigger $t$ is learned via gradient descent to consistently shift embeddings toward $f_{\text{text}}(y_{\text{target}})$ across poisoned samples.

### 3.4 ATTACK IMPACT

We now assess the effectiveness of the backdoor attack in a standard federated prompt-learning setup, where $25\%$ of clients are malicious. These clients inject a learnable noise trigger into a subset of their local data and relabel the triggered samples to a fixed target class. The goal is to induce targeted misclassifications on trigger-inserted test samples, while preserving high performance on clean data.

Table 1: Accuracy with no attack (clean model), clean inputs under attack, and backdoored inputs.

| Dataset | No-Attack | Clean | Backdoor |
|---------|-----------|-------|----------|
| Flowers | 80.9 | 77.9 | 41.7 |
| Pets | 94.5 | 94.2 | 16.3 |
| DTD | 65.2 | 65.6 | 34.8 |
| Aircraft | 32.3 | 32.8 | 93.9 |
| Food101 | 90.7 | 90.0 | 20.6 |

**Backdoor Effectiveness:** Table 1 and Figure 3 show the results of the attack across five datasets. Refer to Appendix C for setup details. We observe that the global model maintains high clean accuracy on all datasets, indicating that benign generalization is largely preserved. At the same time, the backdoor accuracy which is defined as the fraction of trigger-inserted test samples classified as the attacker's target label is significantly elevated, particularly for datasets like FGVC Aircraft (93.9%) and Flowers (41.7%). These results confirm that Federated Prompt Learning systems are vulnerable to backdoor injection even under strong aggregation, and that malicious clients can effectively implant targeted behaviors without degrading global model performance on clean data.

**Comparison with Centralized Backdoor Attacks:** We compare our FL backdoor attack against its centralized counterpart, BadCLIP Bai et al. (2024), which serves as the baseline for prompt-learning backdoor attacks in non-federated settings. BadCLIP achieves near 100% backdoor success by directly poisoning a large portion of the training data and optimizing the trigger in a fully centralized regime. In contrast, our setting uses the standard FedAvg aggregation algorithm and models a more realistic adversary: only a small subset of clients are malicious, and poisoning is confined to local updates. This naturally dilutes the backdoor signal during aggregation and results in lower backdoor accuracy compared to the centralized case. Despite this, our attack achieves high success rates on several datasets, demonstrating that prompt-based FL remains vulnerable even with limited adversarial participation. In Table 1, we report results under the no-defense scenario to highlight how much damage can occur with the default FedAvg setup. We analyze the effectiveness of standard defenses in mitigating this attack later in §5.1.

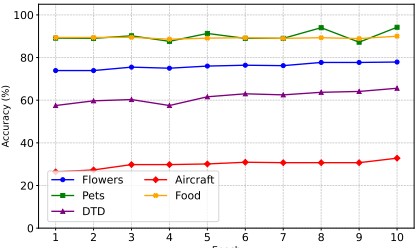

(a) Clean Accuracy

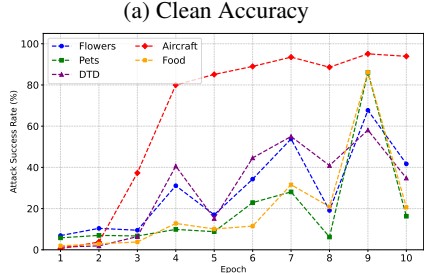

(b) Backdoor Accuracy

Figure 3: Accuracy during attack.

# 4 SABRE-FL: SELECTIVE AND ACCURATE BACKDOOR REJECTION FOR FEDERATED PROMPT LEARNING

Having demonstrated the vulnerability of federated prompt learning to targeted backdoor attacks, we now propose Selective and Accurate Backdoor REjection for Federated Prompt Learning (**SABRE-FL**), a lightweight defense that detects and filters poisoned client updates at the server.

Our key insight is that backdoored inputs induce systematic shifts in the learned representations, as visualized later in §5.2. Even when the trigger is visually imperceptible, it alters the image embedding in a consistent direction enough to cause the downstream model to misclassify the input. This deviation acts as a double-edged sword: it is the very signal that enables the attack, but also the very signal we exploit to build our defense. A similar observation was made in BadCLIP Bai et al. (2024), which showed that the success of prompt-based backdoors arises from consistent embedding-level drift toward the target class. We ask a question: *If this embedding deviation is strong enough to fool the downstream classification model, can it not also be used to detect that the input has been poisoned?*

***Core idea.*** Rather than detecting poisoning in pixel or parameter space, we operate in the embedding space where poisoned examples exhibit a consistent statistical signature. By training a binary classifier on clean and triggered embeddings in an auxiliary setting, we learn to detect this signature.

## 4.1 FORMALIZATION

Let $f_{\text{img}}(\cdot)$ denote the CLIP image encoder. Given a clean input $x$, let $z = f_{\text{img}}(x)$, and for its backdoored version $x^\star = x \oplus t$, let $z^\star = f_{\text{img}}(x^\star)$. Our defense relies on the assumption that backdoored embeddings exhibit a separation margin from the embeddings of the clean ones:

$$\|z - z^\star\|_2 > \epsilon \quad \text{for some } \epsilon > 0 \qquad (5)$$

We simulate this behavior by generating a training dataset $\mathcal{D}_{\text{aux}} = \{(z_i, y_i)\}_{i=1}^N$ of clean and poisoned embeddings on an auxiliary dataset (Caltech-101). Here, $y_i \in \{0, 1\}$ indicates whether $z_i$ is clean or poisoned. We train a detector $D : \mathbb{R}^d \to \{0, 1\}$ by minimizing a standard binary loss:

$$\min_\theta \sum_{i=1}^N \ell(D(z_i; \theta), y_i) \qquad (6)$$

---

**Algorithm 1** SABRE-FL

1: **Pre-train Detector:**
    Generate $\mathcal{D}_{\text{aux}} = \{(z_i, y_i)\}$ from clean/poisoned data
    Train $D : \mathbb{R}^d \to \{0, 1\}$ using cross-entropy
2: **for** each FL round $t = 1$ to $T$ **do**
3:    Server sends prompt $p_{t-1}$ to all clients
4:    **for** each client $C_k$ **do**
5:        **if** malicious **then**
6:            Poison subset: $x^\star = x \oplus t$
7:            Relabel $x^\star \to c_t$, train $p_k$ on poisoned data
8:        **else**
9:            Train $p_k$ on clean data
10:        **end if**
11:        Send $p_k$, embeddings $\{z_j^k\}$ to server
12:    **end for**
13:    **for** each $C_k$ **do**
14:        Compute $S_k = \frac{1}{n_k} \sum_j D(z_j^k)$
15:        Remove top-$m$ clients with highest $S_k$
16:    **end for**
17:    Aggregate accepted $\{p_k\} \to p_t$
18: **end for**

---

**Inference Rule.** At inference time, when a client $C_k$ submits a set of embeddings $\{z_j^k\}_{j=1}^{n_k}$, we compute the mean detector score:

$$S_k = \frac{1}{n_k} \sum_{j=1}^{n_k} D(z_j^k) \qquad (7)$$

Rather than using a fixed threshold $\tau$, we adopt a rank-based heuristic: in each round, the $m$ clients with the highest number of flagged embeddings are excluded from aggregation. This approach assumes an upper bound on the number of malicious clients, consistent with prior work Yin et al. (2018); Shejwalkar & Houmansadr (2021); Fang et al. (2020); Cao et al. (2023); Zhang et al. (2022). More details on client filtering are in Appendix B.

**Lemma.** If a consistent margin $\epsilon$ exists and $D$ achieves zero or near-zero training error on $\mathcal{D}_{\text{aux}}$, then $D$ is expected to generalize well to unseen clients using a noise trigger. This reflects the distributional stability of backdoored embeddings under the frozen encoder.

Table 2: Clean and backdoor accuracy on five datasets. Best backdoor accuracy(lowest) is **bold**.

| Defense | Flowers | | Pets | | DTD | | FGVC Aircraft | | Food101 | |
|---|---|---|---|---|---|---|---|---|---|---|
| | Clean | BD | Clean | BD | Clean | BD | Clean | BD | Clean | BD |
| No Defense | 77.9 | 41.7 | 94.2 | 16.3 | 65.6 | 34.8 | 32.8 | 93.9 | 90.0 | 20.6 |
| Trimmed Mean | 76.8 | 12.3 | 93.7 | 5.6 | 63.7 | 31.0 | 32.4 | 83.1 | 90.0 | 6.4 |
| Median | 77.4 | 10.4 | 94.1 | 5.3 | 65.9 | 28.1 | 32.1 | 79.4 | 90.1 | 5.5 |
| Norm Bounding | 79.0 | 22.0 | 92.6 | 22.5 | 67.6 | 37.5 | 30.9 | 86.2 | 89.7 | 17.2 |
| FLAME | 76.4 | 3.8 | 93.4 | 7.8 | 66.0 | 8.7 | 31.5 | 16.4 | 89.9 | 3.2 |
| SABRE-FL (Ours) | 76.6 | **1.1** | 94.5 | **4.4** | 64.9 | **6.8** | 32.1 | **7.6** | 90.6 | **1.9** |

## 4.2 DETECTOR TRAINING AND DEPLOYMENT

To operationalize the formalization of our defense, we construct an auxiliary training dataset $\mathcal{D}_{\text{aux}} = \{(z_i, y_i)\}_{i=1}^{N}$ composed of CLIP image embeddings and binary labels indicating whether the embedding originates from a clean or poisoned input. To simulate this, we use Caltech-101 as a held-out auxiliary dataset and apply our trigger injection method (Algorithm 1, line 6), to a subset of images to produce poisoned samples. Both clean and triggered images are passed through the frozen image encoder $f_{\text{img}}(\cdot)$ and a fixed prompt learner to obtain their embeddings. These embeddings are then labeled as clean ($y_i = 0$) or poisoned ($y_i = 1$) to construct the training set. We defer the rest of the training details to Appendix C.4.

## 4.3 PRIVACY CONSIDERATIONS

*SABRE-FL operates solely in the embedding space and does not require access to raw data, labels, or gradients. Clients share only CLIP-encoded image representations with the server which are compressed, task-agnostic vectors produced by a frozen backbone.* This strategy is consistent with prior FL paradigms such as vertical FL Liu et al. (2024); Fu et al. (2022); Bai et al. (2023b) and split learning Vepakomma et al. (2018); Thapa et al. (2022), where intermediate features are shared across parties. Moreover, since we use a frozen encoder, the embeddings are less likely to leak private information (more details in Appendix A.4). Unlike gradients or label-conditioned outputs, CLIP embeddings are not trained to retain input-specific details or reconstruct original data. We acknowledge that data extraction attacks are an evolving research concern Carlini et al. (2023); Balle et al. (2022); however, our approach avoids sharing raw data, labels, or gradients, components that are more strongly correlated with reconstruction leakage.

## 5 EXPERIMENTS AND RESULTS

### 5.1 RESULTS

Due to space constraints, we defer setup details and additional experiments to Appendices C & D.

**Effectiveness of SABRE-FL.** We compare our proposed defense, SABRE-FL, to four widely-used robust aggregation techniques: *Trimmed Mean Yin et al. (2018); Xie et al. (2018), Coordinate-wise Median Yin et al. (2018), Norm Bounding Sun et al. (2019)*, and *FLAME Nguyen et al. (2022)*. Results across five datasets are shown in Table 2. Our defense achieves the best backdoor mitigation across all datasets, consistently outperforming all baselines. Notably, SABRE-FL reduces backdoor accuracy to near zero (as low as $1.1\%$ on Flowers and $1.9\%$ on Food101) without degrading clean accuracy. In fact, clean performance remains comparable or superior to baseline methods, highlighting that aggressive filtering of poisoned clients does not impair generalization. While existing methods do reduce the backdoor accuracy relative to the no-defense baseline, they often leave a significant portion of poisoned influence intact, especially on challenging datasets like FGVC Aircraft and DTD. For example, FLAME achieves $16.4\%$ BA on FGVC Aircraft, and Norm Bounding exceeds $30\%$ BA on multiple datasets.

**Robustness and Generalization.** SABRE-FL operates without access to client data distributions or downstream task labels. The detector is trained once on Caltech-101 and generalizes across diverse datasets in our evaluation (e.g., Flowers, DTD, FGVC Aircraft, Food101, Pets). This generalization holds across input domains such as fine-grained object categories (Flowers, Aircraft) and texture-based recognition tasks (DTD), as well as across classification objectives ranging from animal species

Table 3: Backdoor attack effectiveness with and without SABRE-FL at 32-client scale. Each cell shows Clean Accuracy / Backdoor Accuracy (%).

| Dataset | Flowers | Pets | DTD | FGVC Aircraft | Food101 |
|---|---|---|---|---|---|
| No Defense | 74.9 / 43.5 | 88.8 / 25.9 | 59.3 / 46.8 | 29.9 / 89.9 | 89.2 / 32.2 |
| SABRE-FL | 75.0 / **8.5** | 91.1 / **7.2** | 61.0 / **14.1** | 29.7 / **24.7** | 89.7 / **2.8** |

(Pets) to food recognition (Food101). Because the embedding deviation arises from the backdoor mechanism itself, not the specific data distribution, SABRE-FL reliably detects poisoning via a consistent statistical signature in the embedding space. This highlights its robustness across both domains and tasks, making it broadly applicable in real-world federated deployments.

## 5.2 QUALITATIVE ANALYSIS

To demonstrate why our detection mechanism works, we visualize the embeddings of clean images and their poisoned counterparts. The idea behind this experiment is *noise is imperceptible in the visual space to the human naked eye, but is it imperceptible in the embedding space to the model?* This is answered by visualizing the embeddings in a low-dimensional space using a technique like T-SNE Van der Maaten & Hinton (2008). In Figure 4, we show the T-SNE plots for Caltech-101. We show a similar plot in Appendix D.1 for Oxford Flowers. We first train a model with backdoors using the technique similar to BadClip Bai et al. (2024), then we pass clean and noisy images through the image encoder and store the output embeddings. When we plot these embeddings using T-SNE, we can see that there is a clear divide between the features of the clean

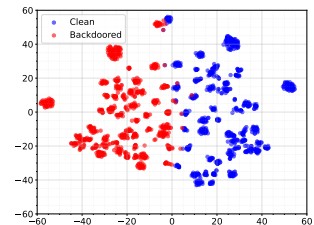

Figure 4: t-SNE visualization on Caltech embeddings. Clean and backdoored samples are clearly separable in the CLIP embedding space.

images and the backdoored images. This validates our intuition behind designing our defense, which lies in the simple fact that if the noise can be used to fool the model into predicting a wrong class, that same noise can also be used to detect if an embedding comes from a clean image or a poisoned one.

## 5.3 ABLATION STUDY

**Impact of Prompt Shot Count:**

The number of shots in prompt learning determines how many samples per class are used to tune the prompt. We study how prompt strength affects both attack success and defense robustness via an ablation over 2, 4, 8, and 16 shots. For each setting, we report clean and backdoor accuracy, with and without our defense, across five benchmark datasets. Figure 5 shows results for DTD; remaining plots are in Appendix D.2.

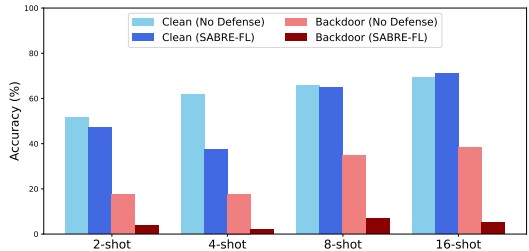

Figure 5: Varying number of shots for DTD

Without any defense, backdoor accuracy increases significantly as the number of shots grows, most notably in datasets like FGVC Aircraft and Food101, where attack success reaches over 85% at 16 shots. This trend suggests that prompt learners become increasingly susceptible to backdoor attacks as they receive more supervision, likely due to stronger memorization of poisoned training samples (more details in Appendix A.2). At the same time, clean accuracy also improves, reflecting the natural benefits of more labeled data. With our defense SABRE-FL enabled, however, backdoor accuracy remains consistently low (under 5%) across all shot counts and datasets. This indicates that our embedding-based detector remains effective even as prompt learners become more expressive. Crucially, clean accuracy under our defense matches or exceeds the no-defense baseline, confirming that the defense does not suppress benign updates. Overall, this experiment highlights that our method provides strong backdoor mitigation without compromising clean performance, even as model capacity increases with additional prompt shots.

**Robustness to Client Scaling:** To evaluate the robustness of SABRE-FL under increased scale, we replicate our backdoor attack and defense experiments with 32 clients. As shown in Table 3, backdoor success rates rise substantially in the absence of defense, reaching 89.9% on FGVC Aircraft and 46.8% on DTD. When SABRE-FL is enabled, backdoor accuracy drops to 24.7% and 14.1%, respectively, demonstrating that our detector remains effective even as the number of participating clients grows. Clean accuracy also remains stable across all datasets, confirming that the defense generalizes to larger federated populations without degrading utility.

**Effect of Malicious Client Proportion:** We vary the proportion of malicious clients on both clean accuracy and backdoor success. In Figure 6, backdoor accuracy rises sharply as the attacker fraction increases. At an attacker rate of 25%, the attack achieves 93.9% success on FGVC and 41.7% on Flowers. At 50% malicious or more, backdoor accuracy exceeds 80% on most datasets and approaches 100% at the highest setting. These results highlight the sensitivity of prompt-based FL to even adversarial participation, especially in few-shot regimes where each client contributes limited data. Notably, clean accuracy remains largely unaffected across all configurations, indicating that the poisoned updates are stealthy and do not visibly degrade global model performance.

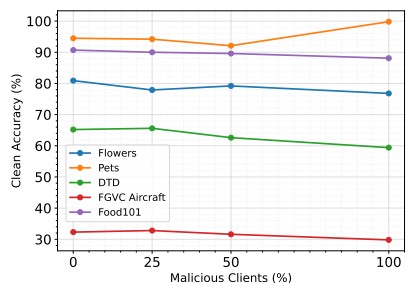

(a) Clean accuracy vs. malicious clients.

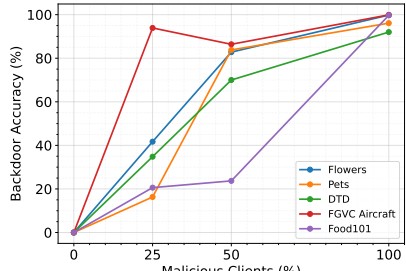

(b) Backdoor accuracy vs. malicious clients.

Figure 6: Effect of increasing malicious client percentage on model performance. Clean accuracy remains stable, while backdoor success increases sharply with more adversarial control.

## 6 LIMITATIONS AND FUTURE WORK

This work brings together three major research areas: federated learning, prompt learning, and backdoor attacks under a unified evaluation framework. Given the breadth of this integration, it is naturally beyond the scope of a single paper to exhaustively explore all possible combinations of settings, attack strategies, and defense variants within this space. Our goal in this paper was to highlight a critical and previously unexamined vulnerability: the susceptibility of federated prompt learning to targeted backdoor attacks. To that end, we carefully selected evaluation settings that isolate this problem and clearly demonstrate both the threat and the effectiveness of SABRE-FL.

However, there are limitations. First, while we focused on data poisoning attacks with learnable triggers, we did not explore model poisoning attacks Fang et al. (2020); Shejwalkar & Houmansadr (2021), where the attacker perturbs client model parameters directly. Future work could compare the relative potency and stealth of model vs. data poisoning in prompt-based FL. Secondly, we used the CLIP ViT-B/16 backbone throughout this study; while it is a representative and widely adopted model, future work may examine other vision-language backbones (e.g., ViT-L, EVA-CLIP, or OpenCLIP variants) to assess generalization across model families.

## 7 CONCLUSION

We show that backdoor attacks are a potent threat to federated prompt learning. We show their success, and use that to design a robust defense, SABRE-FL, against such noise-trigger-based attacks. Our defense is based on the core intuition that the backdoor noise trigger propagates to the embeddings as well. SABRE-FL is a detector model that is able to filter clean and noisy embeddings. Evaluation across five datasets and four baseline defenses shows that our defense outperforms all baselines.

## 8 ACKNOWLEDGEMENTS

We thank the ICLR reviewers for the insightful feedback. This work was supported by NSF award 2452819.

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

# Appendix

We provide additional information for our paper, SABRE-FL: Selective and Accurate Backdoor Rejection for Federated Prompt Learning, in the following order:

- Terminology/Techniques (Appendix A)
- Additional Implementation Details (Appendix B)
- Experimental Setup (Appendix C)
- Additional Results (Appendix D)
- Rebuttal (Appendix E)

## A    TERMINOLOGY/TECHNOLOGIES

### A.1    CLIP: CONTRASTIVE LANGUAGE-IMAGE PRETRAINING

CLIP, short for Contrastive Language-Image Pretraining, is a type of multimodal machine learning model developed by OpenAI Radford et al. (2021). *"Multimodal"* means it can process and relate information from two different types of inputs, in this case, images and natural language. Models like CLIP are referred to as *vision-language models (VLMs)* because they jointly understand both visual and textual information. CLIP was trained on a large dataset of 400 million (image, text) pairs collected from the internet. The idea behind CLIP is simple but powerful: given an image and a sentence, the model learns to tell whether the sentence correctly describes the image. For example, given a photo of a cat and several captions like "a cat," "a dog," or "a painting," CLIP learns to match the correct caption to the image. This is done using a technique called *contrastive learning*, where the model pulls together matching image-text pairs and pushes apart mismatched ones in the embedding space.

CLIP has two components: - An *image encoder* (e.g., a Vision Transformer or ResNet) that converts images into high-dimensional vectors. - A *text encoder* (e.g., a Transformer) that converts sentences into vectors in the same space. After training, CLIP can be used for *zero-shot classification*, where it is given a list of possible text labels and an image, and it predicts which label best matches the image. This makes CLIP very versatile for *downstream tasks*, i.e., tasks that are different from the model's pretraining objective, such as object classification, image retrieval, OCR, or even robotics. During testing, CLIP matches a given test image with the best matching class label (converted into a prompt like "a photo of a class"). In summary, CLIP is a general-purpose vision-language model that learns a shared representation space for images and text without needing explicit labels. It serves as the foundation for prompt learning, which allows users to adapt CLIP to new tasks more effectively.

### A.2    PROMPT LEARNING

A *prompt* is a piece of text that is used to guide a model's predictions. In language models (like GPT), a prompt might be a sentence like "Translate this to French: Hello," and in CLIP, it might be "a photo of a dog." In the original CLIP setup, hand-crafted prompts like "a photo of a class" are used during testing to convert text labels into embeddings. However, *manual prompts are often suboptimal* as they rely on human intuition and may not generalize well across tasks or datasets. This led to the idea of *prompt learning*, where instead of using fixed textual prompts, we learn *soft prompts*, i.e., a set of trainable vectors that replace or augment the context in a prompt. These prompts are optimized during training to improve model performance on a given downstream task.

The pioneering work in this area is *CoOp (Context Optimization)* Zhou et al. (2022b), which introduced learnable prompts for vision-language models like CLIP. In CoOp, the prompt is represented as a series of learnable embeddings $[\boldsymbol{v}_1, \boldsymbol{v}_2, ..., \boldsymbol{v}_N]$, which are prepended to each class name (e.g., "`[v1][v2]...  [vN]` dog") and passed through the text encoder. These prompts are optimized using a small amount of labeled data. Prompt learning has several advantages: (1)It avoids fine-tuning the entire backbone, making it computationally efficient. (2) It adapts the model to new tasks with only a few training examples (few-shot learning). (3) It retains the generalization power of the pretrained model while specializing it for a specific task. Some common *prompt hyperparameters*

include: (1)*Context length (N):* the number of learnable prompt vectors prepended to the class name. (2)*Number of shots:* how many labeled examples per class are used for training. (3)*Class token position:* whether the class label appears at the start, middle, or end of the prompt. Increasing the *number of shots* typically improves accuracy because the model sees more training examples per class, allowing the prompt learner to better capture the features that distinguish different categories. However, prompt learning often performs well even in low-shot settings, making it ideal for domains with limited labeled data.

## A.3 BADCLIP

BadCLIP is a backdoor attack framework proposed in a CVPR 2024 paper Bai et al. (2024), designed to evaluate the vulnerability of prompt-learning-based vision-language models like CLIP. Unlike traditional backdoor attacks that rely on visible patterns or simple data poisoning, BadCLIP crafts *visually imperceptible noise triggers* that manipulate the internal behavior of the model during both training and inference. Similar to CLIP, BadCLIP predicts the correct label by comparing image features to text features derived from prompts (e.g., "a photo of a dog"). In the presence of a backdoor, a small adversarial noise pattern (trigger) is added to the input image. This trigger is optimized during training to cause the image encoder to shift the image embedding closer to the text embedding of an attacker-specified target class (e.g., "cat"), while remaining visually indistinguishable to humans. BadCLIP also adapts the prompt vectors in a *trigger-aware* manner. That is, both image features and context vectors are conditioned on the presence of the backdoor trigger, making the backdoor more robust and more likely to survive training. During inference, even if a clean image is given a trigger, the poisoned model misclassifies it as the target class due to embedding-level drift.

More formally, given a clean image $x$ and a trigger $t$, the backdoored input $x^\star = x \oplus t$ results in an image embedding $f(x^\star)$ that is closer to the prompt-conditioned text embedding of the target class $g(\{V, c_t\})$ than to its true label $g(\{V, c_y\})$. The model predicts the target class $t$ even though the visual appearance corresponds to $y$. BadCLIP is the first backdoor framework using noise-based triggers specifically designed for prompt-tuned CLIP models. Its key insight is that backdoor signals are not limited to the input space but can be embedded into CLIP's latent space, making them both stealthy and effective. SABRE-FL builds on this idea, extending it to the federated learning setting.

## A.4 PRIVACY LEAKAGE

Recent work has demonstrated that it is possible to reconstruct input data from machine learning models Balle et al. (2022); Carlini et al. (2023; 2018). These attacks are known as *reconstruction attacks*. However, such attacks typically require certain strong assumptions. For example, Balle et al. (2022) consider a very strong adversary that knows several data points as well as the weights of the model.

SABRE-FL operates entirely in the representation space of a *frozen CLIP encoder*, meaning the image encoder is never updated with client-specific data. As a result, the embeddings remain generic and task-agnostic, optimized for cross-modal alignment, not input reconstruction. This design choice significantly reduces the risk of privacy leakage, as CLIP embeddings are not trained to retain high-frequency or instance-specific image details.

While representation-level inversion remains an evolving area of research, current attacks often assume more favorable conditions than those present in SABRE-FL. Nevertheless, we acknowledge the broader risk and consider our design to reflect a privacy-utility tradeoff: by accepting lightweight representation sharing with a fixed encoder, we achieve robust backdoor detection without compromising raw inputs or task-specific outputs.

## B    ADDITIONAL IMPLEMENTATION DETAILS

### B.1    DETECTOR THRESHOLDING AND CLIENT FILTERING

In the main paper, we define the detector score $S_k$ for each client $C_k$ as the mean classification output over its submitted embeddings:

$$S_k = \frac{1}{n_k} \sum_{j=1}^{n_k} D(z_j^k)$$

where $D(\cdot)$ is a binary classifier that outputs 1 for poisoned embeddings. While this naturally allows for threshold-based filtering (i.e., flagging clients for which $S_k > \tau$), in practice we adopt a more stable rank-based heuristic.

Specifically, in each communication round, we assume $m$ out of $n$ clients may be malicious, and we remove the $m$ clients with the highest number of flagged embeddings (or highest $S_k$ scores). This avoids the need to hand-tune a static threshold $\tau$ and reflects a standard assumption in robust FL defense literature, where $m$ is typically known or bounded Yin et al. (2018); Xie et al. (2018). This rank-based heuristic is consistent with our earlier detector formulation and preserves the intended semantic interpretation of $S_k$ as a client-level anomaly score.

## C    EXPERIMENTAL SETUP

### C.1    MODEL AND ATTACK SETTINGS

We use the CLIP model in a similar style as that of Bai et. al Bai et al. (2024). ViT-B/16 is used as the image encoder. The pretrained weights are taken from CLIP's released models Radford et al. (2021). We use a context length N of 4, total number of epochs as 10, where 1 is a warmup epoch, and a cosine learning rate scheduler with an initial learning rate of 0.002. Unless specified otherwise, we keep the number of shots to be 8, trigger optimization for 3 epochs, and an SGD optimizer. The maximum noise strength, $\epsilon$, for the backdoor trigger is chosen to be 4. Similar to BadClip, the first class of every dataset is chosen as the target class during the attack.

### C.2    DATASETS

We use datasets that are used in CoOp Zhou et al. (2022b) and BadCLIP Bai et al. (2024). We use the same dataset configuration files they provide. The datasets we use in our experiments are:

- **Caltech-101 Fei-Fei et al. (2004)** is a standard object classification dataset consisting of 9,146 images across 101 object categories and a background class. It has the license CC BY 4.0. Each category contains between 40 and 800 images of objects taken from varying viewpoints and backgrounds. The dataset is known for its moderate intra-class variation and has been widely used in evaluating vision models, especially in low-shot and few-shot learning settings. In our work, we use Caltech-101 as an out-of-distribution (OOD) dataset to train our backdoor detector. Importantly, this dataset is disjoint from the ones used in federated training, allowing us to test whether our detector generalizes across domains.

- **Flowers-102 Nilsback & Zisserman (2008)** is a fine-grained classification dataset consisting of 8,189 images of flowers categorized into 102 species. Each class contains between 40 and 258 samples. The high inter-class similarity and fine-grained nature of the dataset make it a challenging benchmark for vision-language models.

- **The Oxford-IIIT Pets dataset Parkhi et al. (2012)** contains 7,349 images of 37 breeds of cats and dogs. Each class includes approximately 200 images captured in varied poses, lighting conditions, and backgrounds. The dataset presents a mix of inter-class similarity and intra-class diversity, making it suitable for testing the robustness of prompt learners in federated setups. It is available under the license CC BY-SA 4.0.

- **DTD Cimpoi et al. (2014)** (Describable Textures Dataset) is a texture-centric classification dataset with 5,640 images labeled across 47 human-describable texture attributes such as "bumpy," "scaly," or "striped." The dataset emphasizes mid-level visual cues and is used in our evaluation to test

whether prompt-based FL models can maintain robustness when the notion of class is not strictly object-centric.

- **FGVC Aircraft Maji et al. (2013)** contains 10,000 images of 100 aircraft variants grouped by manufacturer and model. It is a fine-grained classification dataset that introduces significant challenges due to subtle inter-class differences and high intra-class consistency. We include it to assess whether backdoor attacks are effective even in domains where prompt learners must capture nuanced visual differences.

- **Food-101 Bossard et al. (2014)** consists of 101,000 images across 101 food categories. The dataset exhibits significant visual diversity, both within and across classes, and is commonly used to benchmark image classification performance under real-world visual noise and clutter. It serves as one of the more large-scale and diverse benchmarks in our federated evaluation.

### C.3 Defense Methods

We compare our technique with four popular defense techniques. Trimmed mean Yin et al. (2018); Xie et al. (2018) is a widely used defense in FL, where the server receives updates from each client, sorts them across each dimension, and then discards the $m$ smallest and lowest values across each dimension. Here, $m$ is the number of malicious clients. Median Yin et al. (2018) is another popular defense mechanism, where the global model is computed by taking the dimension-wise median of the client updates. Norm-bounding Sun et al. (2019) clips the values of client updates to a certain value so they do not exceed that threshold. This threshold is computed by taking the median value of the client updates. FLAME Nguyen et al. (2022) is a more complex defense that first clusters the clients into benign and malicious groups using hdbscan Campello et al. (2013), clips them at a certain threshold, and adds noise to the model parameters to make them resilient to backdoors.

### C.4 Detector Training

We train a detector $D : \mathbb{R}^d \rightarrow \{0, 1\}$ to minimize binary cross-entropy loss over this embedding dataset. The model architecture is a two-layer multilayer perceptron (MLP) with a hidden layer of size 128 and ReLU activation. It takes as input CLIP image embeddings $z_i \in \mathbb{R}^d$ (with $d = 512$) and outputs logits corresponding to the clean or backdoored class. Optimization is performed using the Adam optimizer with a learning rate of $1 \times 10^{-3}$ for 20 epochs, and batch size 64. The detector's file size is a few MBs.

To evaluate cross-domain generalization, we test the trained detector on separate held-out datasets, namely Oxford Flowers, Pets, DTD, FGVC Aircraft, and Food101, each containing a mix of clean and poisoned embeddings. Despite being trained on a single auxiliary dataset, the detector consistently achieves $> 90\%$ accuracy on these unseen domains. This supports our hypothesis that poisoned embeddings exhibit a consistent statistical signature in CLIP space, independent of the underlying dataset or class distribution.

### C.5 Resources

We used PyTorch pytorch for our coding on a Linux-based system. For running experiments, we use our university cluster that has different types of GPUs. Most of our experiments were performed on 12 GB NVIDIA TITANX GPUs. The run time of the experiments depended upon the dataset used, number of shots, and number of clients.

## D Additional Results

### D.1 t-SNE

We show the t-SNE plot of Oxford Flowers clean and backdoored embeddings in Figure 7.

### D.2 Varying Number of Shots

We show the impact of varying the number of shots on all five datasets in Figure 8.

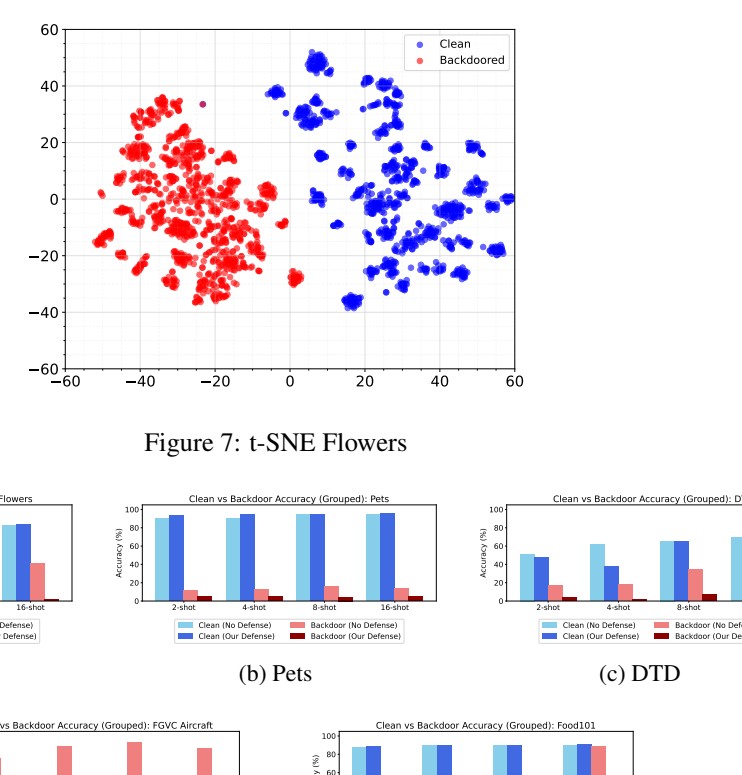

Figure 7: t-SNE Flowers

|  |  |  |
|---|---|---|
| (a) Flowers | (b) Pets | (c) DTD |

|  |  |
|---|---|
| (d) FGVC Aircraft | (e) Food101 |

Figure 8: Our defense consistently reduces backdoor success without degrading clean performance, even as the number of shots increases.

## E   REBUTTAL

In this section, we present new experimental results conducted in response to reviewer feedback. These include visualizations of learned triggers (Figure 9), evaluations under varying data heterogeneity using Dirichlet sampling (Table 4), ablations on trigger strength and optimization steps (Tables 5–6), robustness analysis under imperfect or excessive client filtering (Tables 7–8), and detector generalization across auxiliary datasets (Table 9). Collectively, these results further strengthen our claims regarding the effectiveness, generalizability, and practical robustness of SABRE-FL.

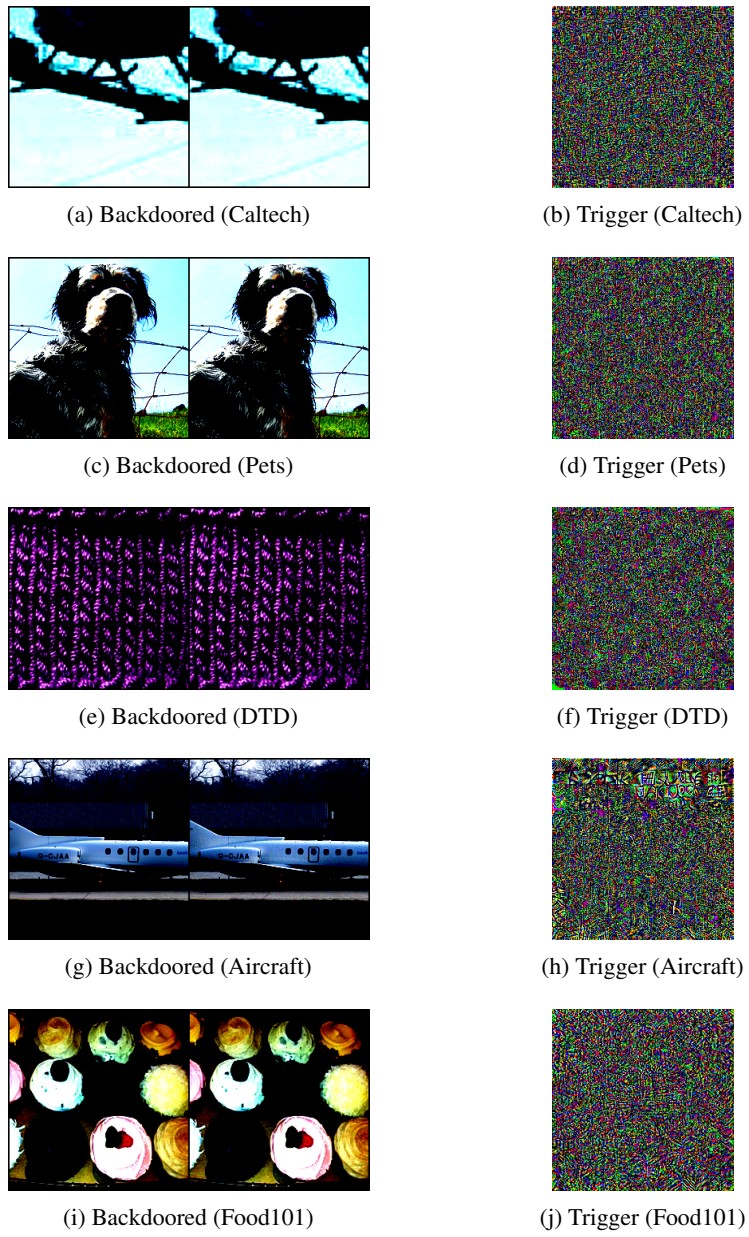

(a) Backdoored (Caltech)      (b) Trigger (Caltech)

(c) Backdoored (Pets)      (d) Trigger (Pets)

(e) Backdoored (DTD)      (f) Trigger (DTD)

(g) Backdoored (Aircraft)      (h) Trigger (Aircraft)

(i) Backdoored (Food101)      (j) Trigger (Food101)

Figure 9: Visual examples of backdoored inputs and corresponding learned triggers. Each row shows a dataset-specific backdoored image (left) and the additive noise trigger alone (right).

Table 4: Clean and backdoor accuracy across varying Dirichlet $\alpha$ values (heterogeneity levels), *with and without SABRE-FL*. Lower $\alpha$ indicates higher non-IID-ness.

| $\alpha$ | Caltech No Defense CA | BA | SABRE-FL CA | BA | Pets No Defense CA | BA | SABRE-FL CA | BA | DTD No Defense CA | BA | SABRE-FL CA | BA | Aircraft No Defense CA | BA | SABRE-FL CA | BA | Food101 No Defense CA | BA | SABRE-FL CA | BA |
|---|---|---|---|---|---|---|---|---|---|---|---|---|---|---|---|---|---|---|---|---|
| 0.9 | 97.7 | 22.3 | 97.7 | **8.4** | 95.2 | 5.8 | 95.3 | **4.6** | 64.9 | 9.3 | 64.5 | **7.1** | 31.9 | 86.7 | 32.5 | **0.1** | 90.0 | 36.0 | 90.4 | **3.0** |
| 0.7 | 97.3 | 22.1 | 97.2 | **8.5** | 94.6 | 10.8 | 94.4 | **4.9** | 61.8 | 11.3 | 64.1 | **9.6** | 30.7 | 85.0 | 30.9 | **33.0** | 90.0 | 14.7 | 90.3 | **2.4** |
| 0.5 | 97.2 | 30.9 | 97.7 | **8.3** | 94.4 | 17.9 | 95.3 | **4.7** | 62.2 | 25.3 | 65.0 | **7.6** | 30.7 | 85.7 | 30.0 | **4.4** | 89.8 | 36.1 | 89.5 | **33.2** |
| 0.3 | 97.5 | 32.6 | 97.4 | **20.5** | 93.0 | 12.2 | 91.0 | **10.0** | 63.5 | 9.6 | 60.2 | **11.0** | 30.3 | 86.7 | 31.5 | **83.9** | 89.7 | 71.6 | 89.2 | **19.9** |
| 0.1 | 97.0 | 24.5 | 96.7 | **13.8** | 94.3 | 12.8 | 92.8 | **7.3** | 60.6 | 12.6 | 59.0 | **11.1** | 31.7 | 81.3 | 30.9 | **89.6** | 89.4 | 56.7 | 89.4 | **37.3** |

Table 5: Effect of trigger strength ($\epsilon$ scaling) and SABRE-FL defense on clean accuracy (CA) and backdoor accuracy (BA). Best BA (lower is better) is in **bold**.

| Setting | Caltech | | Pets | | DTD | | Aircraft | | Food101 | |
|---|---|---|---|---|---|---|---|---|---|---|
| | CA | BA | CA | BA | CA | BA | CA | BA | CA | BA |
| $2 \times \epsilon$ (No Defense) | 97.2 | 51.4 | 93.9 | 6.0 | 64.9 | 51.4 | 31.2 | 92.7 | 90.1 | 33.5 |
| $0.5 \times \epsilon$ (No Defense) | 97.1 | 8.7 | 92.1 | 27.3 | 64.6 | 27.8 | 30.5 | 80.0 | 89.8 | 4.5 |
| $2 \times \epsilon$ + SABRE-FL | 97.1 | 7.6 | 94.6 | 4.6 | 64.4 | 4.6 | 32.0 | 17.2 | 90.7 | 3.9 |
| $0.5 \times \epsilon$ + SABRE-FL | 97.1 | **8.3** | 94.6 | **4.3** | 64.4 | **5.8** | 32.0 | **1.0** | 90.7 | 2.1 |

Table 6: Effect of trigger optimization steps (epochs) on clean accuracy (CA) and backdoor accuracy (BA). Higher CA and lower BA are better.

| Setting | Caltech | | Pets | | DTD | | Aircraft | | Food101 | |
|---|---|---|---|---|---|---|---|---|---|---|
| | CA | BA | CA | BA | CA | BA | CA | BA | CA | BA |
| no defense (1 epoch) | 96.9 | 31.6 | 94.1 | 7.0 | 64.7 | 30.3 | 28.9 | 61.7 | 90.2 | 3.0 |
| no defense (5 epochs) | 97.2 | 58.4 | 94.6 | 6.3 | 61.9 | 41.8 | 31.3 | 87.6 | 90.1 | 22.6 |
| SABRE-FL (1 epoch) | 97.1 | 7.8 | 94.4 | 4.2 | 67.8 | 6.2 | 31.6 | 3.2 | 90.4 | 2.2 |
| SABRE-FL (5 epochs) | 97.3 | 8.1 | 94.7 | 4.3 | 66.2 | 2.7 | 31 | 0 | 90.5 | 3.1 |

Table 7: Effect of imperfect client filtering: we intentionally do *not* remove some malicious clients and evaluate SABRE-FL's robustness on **Pets** and **DTD**. Clean accuracy (CA) drops mildly, while backdoor accuracy (BA) rises with more undetected attackers.

| # Malicious Clients Not Removed | Pets | | DTD | |
|---|---|---|---|---|
| | CA | BA | CA | BA |
| 1 | 92.1 | 4.7 | 62.2 | 11.3 |
| 2 | 91.1 | 7.2 | 61.0 | 14.1 |
| 3 | 90.5 | 9.1 | 61.1 | 19.7 |
| 4 | 89.8 | 10.1 | 60.5 | 25.5 |

Table 8: Effect of over-pruning: SABRE-FL removes the correct number of malicious clients but also accidentally filters out 1–2 benign clients. Despite this, clean accuracy (CA) remains stable and backdoor accuracy (BA) stays low.

| Setting | Caltech | | Pets | | DTD | | Aircraft | | Food101 | |
|---|---|---|---|---|---|---|---|---|---|---|
| | CA | BA | CA | BA | CA | BA | CA | BA | CA | BA |
| 2 Malicious + 1 Benign Removed | 97.3 | 6.6 | 94.8 | 4.5 | 64.5 | 6.8 | 33.4 | 0.4 | 90.5 | 2.0 |
| 2 Malicious + 2 Benign Removed | 97.4 | 7.3 | 94.9 | 4.8 | 63.5 | 5.4 | 31.1 | 17.2 | 90.3 | 2.1 |

Table 9: Evaluating SABRE-FL when the detector is trained on **Flowers** instead of Caltech. Despite the shift in auxiliary dataset, the defense maintains strong generalization across tasks, reducing backdoor accuracy (BA) while preserving clean accuracy (CA).

| Setting | Caltech | | Pets | | DTD | | Aircraft | | Food101 | |
|---|---|---|---|---|---|---|---|---|---|---|
| | CA | BA | CA | BA | CA | BA | CA | BA | CA | BA |
| No Defense | 97.2 | 58.2 | 92.1 | 12.1 | 67.6 | 27.3 | 32.5 | 91.5 | 90.0 | 20.6 |
| SABRE-FL (Trained on Flowers) | 97.1 | **6.1** | 94.6 | **4.4** | 64.4 | **4.7** | 32.0 | **0.2** | 90.7 | **2.2** |

