# OpenReview forum: "SABRE-FL: Selective and Accurate Backdoor Rejection for Federated Prompt Learning"
_ICLR.cc/2026/Conference — ICLR 2026 Poster_

### Official Review · Reviewer_r8mg · 2025-10-27

**Soundness:** 3
**Presentation:** 3
**Contribution:** 3
**Rating:** 6
**Confidence:** 3

**Summary:**

This paper studies the security of Federated Prompt Learning (FPL) and presents SABRE-FL, a defense framework that detects backdoored client updates using embedding-space anomaly detection. The authors first demonstrate that prompt-based federated systems, despite having a smaller attack surface than full-model FL, are highly vulnerable to imperceptible noise-trigger attacks. SABRE-FL then leverages CLIP embedding deviations to filter malicious updates without accessing raw client data or labels.

**Strengths:**

1. Novel and timely problem. The problem of backdoor attacks in federated prompt learning is novel and interesting. The paper fills an important gap in understanding the security risks of adapting foundation models via prompt learning in a decentralized setting.
2. Clear intuition and methodology. The core idea is intuitive and well-motivated. The defense operates in embedding space, aligning with the privacy constraints of FL.
3. Strong empirical evaluation. Extensive experiments across five datasets and four defense baselines. SABRE-FL achieves the lowest backdoor accuracy while maintaining high clean accuracy.

**Weaknesses:**

1. Limited threat model. Only data poisoning is considered. Model poisoning or adaptive strategies are not analyzed. It is unclear whether an attacker aware of SABRE-FL could evade embedding-space detection. The assumption that the attacker controls 25% of all clients is generally considered to be high compared with existing literatures.
2. Detector training assumptions. The detector is trained using poisoned embeddings generated on an auxiliary dataset. It is not obvious how a real-world server would obtain such poisoned examples to train the detector. The paper should justify the practicality of this pre-training phase.
3. More explanation on comparison to FLAME and other baselines. FLAME also uses embedding-based filtering. The paper needs a clearer conceptual distinction and justification of why SABRE-FL is fundamentally stronger.

**Questions:**

1. Could authors provide the effectiveness of the method against adaptive attackers?
2. Could authors justify the practicality of the pretraining phase?

---

> ### Author Response · Authors · 2025-11-20
> **Rebuttal by Authors**
>
> We thank the reviewer for recognizing the novelty of studying backdoor threats in Federated Prompt Learning and for highlighting the clarity and strength of our methodology and experiments. Below, we address each concern in turn.
>
> ## Adaptive Attacker Robustness
> We agree that evaluating SABRE-FL against adaptive attackers is an important direction for future work. While our current setup assumes a fixed attack strategy, we provide insight into the defense’s resilience under partial evasion through client filtering ablations (Appendix F, Tables 7–8). Specifically, we simulate (1) underestimation, where a stealthy attacker avoids detection and remains in aggregation, and (2) overestimation, where benign clients are mistakenly removed. Results show that even in these imperfect scenarios, SABRE-FL maintains high clean accuracy and significantly suppresses backdoor success. These findings suggest that the detector is reasonably robust to adaptive behavior that might evade direct flagging. We will expand on this in future work with explicit adaptive attack designs and model poisoning as well.
>
> **Table: Effect of imperfect client filtering (Table 7, Appendix F)**
> We intentionally do *not* remove some malicious clients and evaluate SABRE-FL’s robustness on **Pets** and **DTD**. Clean accuracy (CA) drops mildly, while backdoor accuracy (BA) rises with more undetected attackers.
>
> | # Malicious Clients Not Removed | Pets CA | Pets BA | DTD CA | DTD BA |
> |----------------------------------|---------|---------|--------|--------|
> | 1                                | 92.1    | 4.7     | 62.2   | 11.3   |
> | 2                                | 91.1    | 7.2     | 61.0   | 14.1   |
> | 3                                | 90.5    | 9.1     | 61.1   | 19.7   |
> | 4                                | 89.8    | 10.1    | 60.5   | 25.5   |
>
> **Table: Overestimation, i.e., removing benign clients too (Table 8, Appendix F)**
> | Setting                         | Caltech CA | Caltech BA | Pets CA | Pets BA | DTD CA | DTD BA | Aircraft CA | Aircraft BA | Food101 CA | Food101 BA |
> |---------------------------------|------------|------------|---------|---------|--------|--------|-------------|-------------|------------|-------------|
> | 2 Malicious + 1 Benign Removed  | 97.3       | 6.6        | 94.8    | 4.5     | 64.5   | 6.8    | 33.4        | 0.4         | 90.5       | 2.0         |
> | 2 Malicious + 2 Benign Removed  | 97.4       | 7.3        | 94.9    | 4.8     | 63.5   | 5.4    | 31.1        | 17.2        | 90.3       | 2.1         |
>
>
>
>
>
> ## Justifying the pre-training phase
> Thank you for raising this important point. In many real-world FL security systems, the server proactively prepares for anticipated threats (e.g., model poisoning, backdoors) using available auxiliary resources. Similarly, SABRE-FL assumes the server has access to an out-of-distribution (OOD) dataset, publicly available or synthetic, which is a common assumption in FL defenses and OOD detection literature. Poisoned embeddings can be synthetically generated by applying a generic input-space trigger to this auxiliary dataset using the same encoder architecture. Clean and poisoned embeddings are then used to train a neural network to detect statistical deviations induced by the backdoor. This process is lightweight, privacy-preserving, and attacker-agnostic, and does not require access to client data or any specific knowledge of the deployed attack.
>
> ## Detector training practicality
> We agree that practical feasibility is important. In many FL settings, it is standard to assume access to public or auxiliary data for server-side calibration, common in OOD detection and FL defense works (e.g., FLTrust, FLDetector). Our detector uses such auxiliary data (Caltech-101) to simulate poisoned embeddings offline. Crucially, this does not require adversary knowledge: the trigger is synthetic and generic, and the goal is to capture embedding deviations induced by misalignment with class semantics. As shown in Table 9 (Appendix F), even when trained on a different auxiliary dataset (e.g., Flowers), the detector generalizes well, suggesting the learned decision boundary is dataset-agnostic.
>
> ## Comparison to FLAME
> We appreciate the reviewer highlighting this connection. FLAME is a strong defense for full-model FL, leveraging unsupervised clustering and perturbation-based heuristics to mitigate backdoors. However, SABRE-FL adopts a different approach tailored specifically to prompt-based FL: it uses a supervised binary detector trained on auxiliary data to capture embedding-level deviations caused by backdoor triggers. This allows for more fine-grained, client-level attribution rather than coarse cluster-level decisions. Additionally, FLAME was originally developed for settings with dynamic encoder updates, while SABRE-FL is designed for the frozen-encoder, prompt-only paradigm, offering a lightweight and modular defense well-suited to modern vision-language FL deployments.

---

> > ### Comment · Reviewer_r8mg · 2025-11-27
> >
> > Thanks for providing additional experiment results and the detailed response. I choose to keep my initial score.

---

### Official Review · Reviewer_4QUt · 2025-10-30

**Soundness:** 3
**Presentation:** 3
**Contribution:** 3
**Rating:** 6
**Confidence:** 3

**Summary:**

his paper proposes a method named SABRE-FL to defend Federated Prompt Learning (FPL) against backdoor attacks. It takes the form of a server-side defense mechanism that uses a lightweight detector to filter malicious client updates. To achieve this, they employ an anomaly detector that operates in the CLIP embedding space, which is trained offline on out-of-distribution (OOD) data. The paper outlines the vulnerability of FPL to malicious clients injecting learnable noise triggers and proposes a solution to mitigate these issues. Their main contribution is SABRE-FL, a lightweight and modular defense framework, which leverages the consistent deviation produced by poisoned samples in the embedding space to identify and filter poisoned prompt updates. They conducted an empirical study of the attack across five diverse datasets, concluding that FPL is highly vulnerable to backdoor attacks while still maintaining high clean data accuracy. Finally, an empirical study of the defense was conducted, concluding that SABRE-FL demonstrates superior performance compared to four other baseline defenses, as it can reduce backdoor accuracy while preserving clean accuracy.

**Strengths:**

1.This work presents the first systematic study of backdoor attack vulnerabilities within the Federated Prompt Learning (FPL) paradigm. This exploratory contribution is significant as it illuminates a critical and previously unexamined attack dimension.
2.The paper proposes SABRE-FL, a novel server-side defense mechanism. The core of this mechanism involves using a lightweight MLP, trained offline on an out-of-distribution (OOD) dataset, to detect embedding-space anomalies.
3.A key advantage of this defense is its generalizability. The experiments demonstrate that the detector, trained on a single auxiliary dataset, can effectively generalize and be applied across five other distinct task datasets.

**Weaknesses:**

1.SABRE-FL is essentially an anomaly detector. In heterogeneous (Non-IID) FL scenarios, natural shifts in data distribution are an inherent characteristic. The paper provides no evidence that the detector D can distinguish between malicious offsets caused by the attack and benign shifts arising from this data heterogeneity. This casts serious doubt on the method's effectiveness in realistic FL settings.
2.The defense mechanism relies on an assumption that is difficult to satisfy in practice: the server must know the exact number of malicious clients m,a prior in each round to filter out the clients with the top-m highest scores. This assumption is unrealistic for most real-world scenarios.
3.SABRE-FL (according to Algorithm 1) requires clients to upload the embeddings for all their local data in each round. This is likely to incur substantial communication overhead.
4.The paper only provides a small-scale experiment (32 clients) in Appendix E.3 and lacks an evaluation on larger-scale federated networks.

**Questions:**

1.If m is unknown or mis-specified (e.g., if the server underestimates the number of attackers), how does the defense performance of SABRE-FL (both CA and BA) degrade?
2.I would like to see some discussion regarding the communication overhead introduced by SABRE-FL.
3.Could the authors provide more insight into the results on the FGVC Aircraft dataset (Figure 8d)? In the 'no defense' setting, this dataset exhibits two extremes: the lowest Clean Accuracy (CA) and the highest Backdoor Accuracy (BA), and these results differ significantly from those of the other datasets.

---

> ### Author Response · Authors · 2025-11-20
> **Rebuttal by Authors (Part 1/2)**
>
> We sincerely thank the reviewer for highlighting the novelty of our exploration of backdoor threats in Federated Prompt Learning and the generalizability of our SABRE-FL defense. Below, we address each concern and clarify our design choices.
>
> ## Robustness under Non-IID settings:
> We agree that realistic federated learning deployments must account for varying levels of client heterogeneity. In response, we conducted a comprehensive set of experiments using Dirichlet-based data partitioning with α ∈ {0.9, 0.7, 0.5, 0.3, 0.1} to simulate increasing degrees of non-IID distribution shift. As α decreases, data becomes more skewed across clients. Results (see Table4 and AppendixF) show that SABRE-FL remains robust under mild to moderate heterogeneity, maintaining strong clean accuracy and significantly reducing backdoor success. While performance degrades slightly at extreme heterogeneity (e.g., α = 0.1), the defense continues to provide substantial protection. These findings support our core hypothesis: the embedding deviations introduced by backdoor triggers remain statistically separable from benign variations induced by data heterogeneity.
>
> **Table: Effect of varying Dirichlet Distribution, i.e., non-IID-ness (Table 4, Appendix F)**
>
> **_Due to space-constraints, Caltech is written as Calt. Aircraft is written as Airc. No-Defense is ND. Sabre-FL is SF_.**
> | $\alpha$ |   |   |   | **Calt** |      |   |      |      |   |   |   | **Pets** |      |   |      |      |   |   |   | **DTD** |      |   |      |      |   |   |   | **Airc** |      |   |      |      |   |   |   | **Food** |      |      |      |
> |----------|---|---|---|----------|------|---|------|------|---|---|---|----------|------|---|------|------|---|---|---|---------|------|---|------|------|---|---|---|----------|------|---|------|------|---|---|---|----------|------|------|------|
> |          |   |   |   | ND       |      |   | SF   |      |   |   |   | ND       |      |   | SF   |      |   |   |   | ND      |      |   | SF   |      |   |   |   | ND       |      |   | SF   |      |   |   |   | ND       |      | SF   |      |
> |          |   |   |   | CA       | BA   |   | CA   | BA   |   |   |   | CA       | BA   |   | CA   | BA   |   |   |   | CA      | BA   |   | CA   | BA   |   |   |   | CA       | BA   |   | CA   | BA   |   |   |   | CA       | BA   | CA   | BA   |
> | 0.9      |   |   |   | 97.7     | 22.3 |   | 97.7 | 8.4  |   |   |   | 95.2     | 5.8  |   | 95.3 | 4.6  |   |   |   | 64.9    | 9.3  |   | 64.5 | 7.1  |   |   |   | 31.9     | 86.7 |   | 32.5 | 0.1  |   |   |   | 90.0     | 36.0 | 90.4 | 3.0  |
> | 0.7      |   |   |   | 97.3     | 22.1 |   | 97.2 | 8.5  |   |   |   | 94.6     | 10.8 |   | 94.4 | 4.9  |   |   |   | 61.8    | 11.3 |   | 64.1 | 9.6  |   |   |   | 30.7     | 85.0 |   | 30.9 | 33.0 |   |   |   | 90.0     | 14.7 | 90.3 | 2.4  |
> | 0.5      |   |   |   | 97.2     | 30.9 |   | 97.7 | 8.3  |   |   |   | 94.4     | 17.9 |   | 95.3 | 4.7  |   |   |   | 62.2    | 25.3 |   | 65.0 | 7.6  |   |   |   | 30.7     | 85.7 |   | 30.0 | 4.4  |   |   |   | 89.8     | 36.1 | 89.5 | 33.2 |
> | 0.3      |   |   |   | 97.5     | 32.6 |   | 97.4 | 20.5 |   |   |   | 93.0     | 12.2 |   | 91.0 | 10.0 |   |   |   | 63.5    | 9.6  |   | 60.2 | 11.0 |   |   |   | 30.3     | 86.7 |   | 31.5 | 83.9 |   |   |   | 89.7     | 71.6 | 89.2 | 19.9 |
> | 0.1      |   |   |   | 97.0     | 24.5 |   | 96.7 | 13.8 |   |   |   | 94.3     | 12.8 |   | 92.8 | 7.3  |   |   |   | 60.6    | 12.6 |   | 59.0 | 11.1 |   |   |   | 31.7     | 81.3 |   | 30.9 | 89.6 |   |   |   | 89.4     | 56.7 | 89.4 | 37.3 |

---

> ### Author Response · Authors · 2025-11-20
> **Rebuttal by Authors (Part 2/2)**
>
> ## Unknown or misestimated $m$:
> We appreciate the reviewer’s concern. SABRE-FL follows the classical FL defense setup, where the server removes the top-m suspicious clients based on a known or estimated attacker budget, an assumption also used by Trimmed Mean, Median, and Krum. However, we agree that the performance under imperfect estimates of m is a critical practical concern. To this end, we present a comprehensive sensitivity analysis in Appendix F (Tables 7 and 8) that evaluates both underestimation (failing to remove up to 4 malicious clients) and overestimation (erroneously removing up to 2 benign clients). Results show that SABRE-FL remains robust in both settings: clean accuracy remains stable under overestimation, while backdoor accuracy rises gradually under underestimation. This suggests SABRE-FL degrades gracefully even when m is not precisely known.
>
> **Table: Effect of imperfect client filtering (Table 7, Appendix F)**
> We intentionally do *not* remove some malicious clients and evaluate SABRE-FL’s robustness on **Pets** and **DTD**. Clean accuracy (CA) drops mildly, while backdoor accuracy (BA) rises with more undetected attackers.
>
> | # Malicious Clients Not Removed | Pets CA | Pets BA | DTD CA | DTD BA |
> |----------------------------------|---------|---------|--------|--------|
> | 1                                | 92.1    | 4.7     | 62.2   | 11.3   |
> | 2                                | 91.1    | 7.2     | 61.0   | 14.1   |
> | 3                                | 90.5    | 9.1     | 61.1   | 19.7   |
> | 4                                | 89.8    | 10.1    | 60.5   | 25.5   |
>
> **Table: Overestimation, i.e., removing benign clients too (Table 8, Appendix F)**
> | Setting                         | Caltech CA | Caltech BA | Pets CA | Pets BA | DTD CA | DTD BA | Aircraft CA | Aircraft BA | Food101 CA | Food101 BA |
> |---------------------------------|------------|------------|---------|---------|--------|--------|-------------|-------------|------------|-------------|
> | 2 Malicious + 1 Benign Removed  | 97.3       | 6.6        | 94.8    | 4.5     | 64.5   | 6.8    | 33.4        | 0.4         | 90.5       | 2.0         |
> | 2 Malicious + 2 Benign Removed  | 97.4       | 7.3        | 94.9    | 4.8     | 63.5   | 5.4    | 31.1        | 17.2        | 90.3       | 2.1         |
>
>
> ## Communication overhead from uploading embeddings:
> SABRE-FL requires each client to send only a 32×512 FP16 embedding matrix per round, corresponding to 32-shot prompt tuning. This amounts to just 32 KB per client per round. Even in large-scale FL settings, the total overhead remains modest: for example, 10 clients → 320KB, 100 clients → 3.2MB, and 1000 clients → 31MB per round. Compared to full model uploads, SABRE-FL’s overhead is lightweight and scalable.
>
> ## Small-scale FL evaluation:
> We used 32 clients in line with prior prompt-based FL works (e.g., PromptFL, 2023). However, we agree with the reviewer that scaling to a very large scale is important and will provide us with useful insights. Due to time and computational constraints, we are unable to perform that study right now. We highlight this as a future direction.
>
> ## Clarification on FGVC Aircraft dataset (Figure 8d):
> FGVC Aircraft contains fine-grained classes with subtle visual differences, making it more vulnerable to embedding manipulation. Triggered inputs in this dataset shift more strongly toward the target class in CLIP space, leading to high backdoor accuracy. At the same time, the baseline accuracy is low due to the inherent difficulty of the task. This demonstrates that prompt-based FL is especially fragile under fine-grained domains.

---

### Official Review · Reviewer_jAGJ · 2025-11-01

**Soundness:** 3
**Presentation:** 4
**Contribution:** 3
**Rating:** 6
**Confidence:** 3

**Summary:**

The paper investigates the security vulnerabilities of Federated Prompt Learning (FPL) under backdoor attacks. The authors show that FPL models can be compromised when malicious clients inject visually imperceptible, learnable noise triggers into images, leading to targeted misclassification while maintaining high clean accuracy. To counter this, they propose SABRE-FL, a modular and lightweight server-side defense that employs an embedding-space anomaly detector—trained offline on out-of-distribution (OOD) data—to identify and filter poisoned prompt updates without accessing raw client data or labels. Experiments demonstrate that SABRE-FL significantly reduces backdoor accuracy while preserving clean performance across various datasets and FPL settings.

**Strengths:**

- Pioneers the study of backdoor threats in the emerging FPL paradigm.
- Introduces a well-motivated and FPL-specific backdoor mechanism based on learnable, imperceptible noise triggers.
- Clear writing and strong organization, aided by effective visual explanations of both attack and defense designs.

**Weaknesses:**

1. The paper claims the noise triggers are *visually imperceptible*, but lacks direct image comparisons. Including visual examples (original vs. triggered) or a qualitative study would strengthen this claim.
2. SABRE-FL removes the top-*m* suspicious clients, assuming *m* is known. An analysis of sensitivity to inaccurate estimates of *m* would clarify robustness in real-world settings.
3. The distinction between the proposed attack and a federated adaptation of BadCLIP should be elaborated—what novel properties or mechanisms are introduced?
4. The effect of data heterogeneity (Non-IID settings) on SABRE-FL’s detection performance is not well-studied; benign diversity may confound the embedding-based detector.
5. SABRE-FL’s effectiveness depends on the defender’s ability to model known trigger behaviors when training its detector offline. However, this reliance makes it vulnerable to novel or unconventional backdoor types—such as semantic, geometric, or other model-poisoning attacks—that fall outside the learned embedding distribution.
6. Additional comparisons to recent strong baselines such as **Deepsight** [1] or **BackdoorIndicator** [2] would better contextualize SABRE-FL’s improvements.

[1]. Rieger, Phillip, et al. "Deepsight: Mitigating backdoor attacks in federated learning through deep model inspection." arXiv preprint arXiv:2201.00763 (2022).

[2]. Li, Songze, and Yanbo Dai. "{BackdoorIndicator}: Leveraging {OOD} Data for Proactive Backdoor Detection in Federated Learning." 33rd USENIX Security Symposium (USENIX Security 24). 2024.

**Questions:**

Please address the aforementioned weaknesses, particularly by including qualitative visualizations, sensitivity analyses for m, comparisons to stronger defenses, and discussions of detector robustness under Non-IID and adaptive attack scenarios.

---

> ### Author Response · Authors · 2025-11-20
> **Rebuttal by Authors (Part 1/2)**
>
> We thank the reviewer for the positive assessment of our paper’s novelty, writing quality, and clear presentation. We address each concern in detail below.
>
> ## Visualizing Trigger Imperceptibility
> We appreciate the reviewer’s suggestion. In response, we have added visualizations of clean, triggered, and backdoored samples from each dataset in **Appendix F (Figure 9)**. These examples confirm that the learned triggers remain visually subtle, often imperceptible to the human eye, while still successfully altering the model's behavior. We believe this strengthens the qualitative understanding of the attack’s stealthiness, and we plan to expand this analysis further in the final version.
>
> ## Sensitivity to \$m\$ (malicious client count)
> Thank you for raising this concern. In our original setup, SABRE-FL assumes a known attacker budget $m$, consistent with standard FL defenses such as Trimmed Mean and Median, which remove the top-$m$ outlier clients. To evaluate robustness under misestimation, we now include experiments that (i) underestimate $m$ by failing to remove up to 4 malicious clients, and (ii) overestimate $m$ by additionally removing 1–2 benign clients. As shown in Appendix F, Tables 7–8, SABRE-FL continues to preserve clean accuracy and suppress backdoor accuracy under both scenarios, indicating resilience to imperfect client filtering.
>
>
> **Table: Effect of imperfect client filtering (Table 7, Appendix F)**
> We intentionally do *not* remove some malicious clients and evaluate SABRE-FL’s robustness on **Pets** and **DTD**. Clean accuracy (CA) drops mildly, while backdoor accuracy (BA) rises with more undetected attackers.
>
> | # Malicious Clients Not Removed | Pets CA | Pets BA | DTD CA | DTD BA |
> |----------------------------------|---------|---------|--------|--------|
> | 1                                | 92.1    | 4.7     | 62.2   | 11.3   |
> | 2                                | 91.1    | 7.2     | 61.0   | 14.1   |
> | 3                                | 90.5    | 9.1     | 61.1   | 19.7   |
> | 4                                | 89.8    | 10.1    | 60.5   | 25.5   |
>
> **Table: Overestimation, i.e., removing benign clients too (Table 8, Appendix F)**
> | Setting                         | Caltech CA | Caltech BA | Pets CA | Pets BA | DTD CA | DTD BA | Aircraft CA | Aircraft BA | Food101 CA | Food101 BA |
> |---------------------------------|------------|------------|---------|---------|--------|--------|-------------|-------------|------------|-------------|
> | 2 Malicious + 1 Benign Removed  | 97.3       | 6.6        | 94.8    | 4.5     | 64.5   | 6.8    | 33.4        | 0.4         | 90.5       | 2.0         |
> | 2 Malicious + 2 Benign Removed  | 97.4       | 7.3        | 94.9    | 4.8     | 63.5   | 5.4    | 31.1        | 17.2        | 90.3       | 2.1         |
>
>
> ## Distinction from BadCLIP
> While we draw inspiration from BadCLIP [Bai et al., CVPR 2024], our adaptation to the federated setting introduces new challenges and design decisions:
> * **Locality constraint:** Each malicious client optimizes its own private trigger, rather than a single global one.
> * **Prompt-only training:** The attack must work through prompt vector updates without modifying the frozen backbone.
> * **Aggregation dilution:** In FL, poisoned prompt vectors are diluted via aggregation, weakening backdoor signals and requiring stronger trigger embedding alignment.
> * **Data Distribution Shift:** Centralized attacks assume access to a unified, i.i.d. dataset. In federated learning, client datasets are non-i.i.d. and disjoint, meaning the backdoor trigger must be robust to unseen client distributions and class co-occurrence patterns, which increases the difficulty of effective poisoning.

---

> ### Author Response · Authors · 2025-11-20
> **Rebuttal by Authors (Part 2/2)**
>
> ## Effect of Non-IID data
> We appreciate this point and agree it is crucial for real-world applicability. In response, we conducted additional experiments using Dirichlet-based partitioning to simulate increasing data heterogeneity across clients (Dirichlet α ∈ {0.9, 0.7, 0.5, 0.3, 0.1}). As shown in Appendix F, Table 4, while extreme heterogeneity (low α) does slightly degrade defense performance, SABRE-FL remains largely effective at filtering poisoned updates. These findings suggest that the embedding deviations introduced by backdoor triggers remain distinguishable from benign shifts due to natural heterogeneity.
>
> **Table: Effect of varying Dirichlet Distribution, i.e., non-IID-ness (Table 4, Appendix F)**
>
> **_Due to space-constraints, Caltech is written as Calt. Aircraft is written as Airc. No-Defense is ND. Sabre-FL is SF_.**
>
> | $\alpha$ |   |   |   | **Calt** |      |   |      |      |   |   |   | **Pets** |      |   |      |      |   |   |   | **DTD** |      |   |      |      |   |   |   | **Airc** |      |   |      |      |   |   |   | **Food** |      |      |      |
> |----------|---|---|---|----------|------|---|------|------|---|---|---|----------|------|---|------|------|---|---|---|---------|------|---|------|------|---|---|---|----------|------|---|------|------|---|---|---|----------|------|------|------|
> |          |   |   |   | ND       |      |   | SF   |      |   |   |   | ND       |      |   | SF   |      |   |   |   | ND      |      |   | SF   |      |   |   |   | ND       |      |   | SF   |      |   |   |   | ND       |      | SF   |      |
> |          |   |   |   | CA       | BA   |   | CA   | BA   |   |   |   | CA       | BA   |   | CA   | BA   |   |   |   | CA      | BA   |   | CA   | BA   |   |   |   | CA       | BA   |   | CA   | BA   |   |   |   | CA       | BA   | CA   | BA   |
> | 0.9      |   |   |   | 97.7     | 22.3 |   | 97.7 | 8.4  |   |   |   | 95.2     | 5.8  |   | 95.3 | 4.6  |   |   |   | 64.9    | 9.3  |   | 64.5 | 7.1  |   |   |   | 31.9     | 86.7 |   | 32.5 | 0.1  |   |   |   | 90.0     | 36.0 | 90.4 | 3.0  |
> | 0.7      |   |   |   | 97.3     | 22.1 |   | 97.2 | 8.5  |   |   |   | 94.6     | 10.8 |   | 94.4 | 4.9  |   |   |   | 61.8    | 11.3 |   | 64.1 | 9.6  |   |   |   | 30.7     | 85.0 |   | 30.9 | 33.0 |   |   |   | 90.0     | 14.7 | 90.3 | 2.4  |
> | 0.5      |   |   |   | 97.2     | 30.9 |   | 97.7 | 8.3  |   |   |   | 94.4     | 17.9 |   | 95.3 | 4.7  |   |   |   | 62.2    | 25.3 |   | 65.0 | 7.6  |   |   |   | 30.7     | 85.7 |   | 30.0 | 4.4  |   |   |   | 89.8     | 36.1 | 89.5 | 33.2 |
> | 0.3      |   |   |   | 97.5     | 32.6 |   | 97.4 | 20.5 |   |   |   | 93.0     | 12.2 |   | 91.0 | 10.0 |   |   |   | 63.5    | 9.6  |   | 60.2 | 11.0 |   |   |   | 30.3     | 86.7 |   | 31.5 | 83.9 |   |   |   | 89.7     | 71.6 | 89.2 | 19.9 |
> | 0.1      |   |   |   | 97.0     | 24.5 |   | 96.7 | 13.8 |   |   |   | 94.3     | 12.8 |   | 92.8 | 7.3  |   |   |   | 60.6    | 12.6 |   | 59.0 | 11.1 |   |   |   | 31.7     | 81.3 |   | 30.9 | 89.6 |   |   |   | 89.4     | 56.7 | 89.4 | 37.3 |
>
> ## Vulnerability to novel backdoor types
> We agree that evaluating SABRE-FL under broader threat models, including semantic, geometric, or model-poisoning attacks, is an important direction for future work. While our current focus is on input-space trigger detection in prompt-based FL, SABRE-FL operates purely in the embedding space and is agnostic to visual features or class labels. This design makes it a flexible first-line defense that can generalize beyond the exact trigger distribution used during training, though stronger or specialized defenses may be needed for fundamentally different attack modalities. We will update our limitations and future work text to reflect this point.
>
> ## Additional comparisons
> Thank you for highlighting these relevant works. While we were unable to conduct additional baseline comparisons due to computational constraints, we provide a conceptual distinction here. Deepsight introduces fine-grained model inspection and clustering techniques to detect poisoned updates, while BackdoorIndicator leverages out-of-distribution (OOD) tasks to proactively detect backdoors via server-side interventions. In contrast, SABRE-FL focuses on detecting backdoor-aligned deviations in embedding space using a lightweight binary detector trained offline. Rather than clustering or task injection, our method relies on the statistical regularity of poisoned representation patterns and is tailored to prompt-based federated settings. We view SABRE-FL as complementary to these approaches and addressing a distinct setting in the growing landscape of FL defenses. We will consider this  comparison as an important future direction and update in the final version of our paper.

---

> > ### Comment · Reviewer_jAGJ · 2025-11-25
> >
> > Thank you to the authors for being highly responsive and for investing significant effort in addressing my concerns through additional experiments, qualitative visualizations, and clarifying discussions. Despite these improvements, my rating remains at 6.

---

### Official Review · Reviewer_D5Ed · 2025-11-02

**Soundness:** 2
**Presentation:** 2
**Contribution:** 2
**Rating:** 2
**Confidence:** 3

**Summary:**

This paper studies backdoor attacks in Federated Prompt Learning (FPL) and introduces SABRE-FL, a defense framework that detects and filters poisoned client updates using representation-space anomaly detection. The key idea is to train a binary detector on CLIP embeddings from an auxiliary dataset which includes clean and triggered/poisoned samples, so the model can identify abnormal embedding patterns corresponding to malicious clients' updates during FL aggregation.

The method is evaluated on five datasets, which are Flowers, Pets, DTD, FGVC Aircraft, and Food101 under varying malicious client ratios, showing that SABRE-FL achieves low backdoor success rates while preserving clean accuracy.

**Strengths:**

1. This paper studies backdoor attacks in federated prompt learning (FPL), where only prompt parameters, not full model weights, are shared; this is timely and relevant as CLIP-style adaptations in FL are increasingly used.
2. The method shows potential generalizability: a detector trained on Caltech-101 transfers to datasets not seen during training.
3. The paper provides clear motivation and visualization that support the defense’s intuition; the results look promising in tackling the backdoor attack tested.

**Weaknesses:**

1. The methodology section lacks important details.
(i) The paper describes the trigger as a learnable noise pattern but does not explain how it is optimized, what loss function or parameters are used, and how it interacts with the local prompt updates (e.g., whether it uses SGD, PGD, or another generator).
(ii) The defense critically depends on the parameter $m$—the number of clients excluded from aggregation each round—but there is no principled method or empirical guideline for setting this value.
(iii) The detector is trained on an auxiliary dataset (Caltech-101) with synthetically poisoned samples, yet the paper provides little justification that this dataset captures the diversity of real backdoor triggers or resembles the adversary’s trigger design. The assumption that embedding deviations generalize across datasets and trigger types is unverified.

2. Several important ablation studies are missing, such as varying $m$, testing under different non-IID settings, changing the auxiliary dataset, and exploring the effects of trigger strength, magnitude, and optimization steps.

3. The paper does not visualize or analyze the learned trigger patterns that drive the backdoor. Showing how the trigger alters image embeddings or prediction confidence would make the mechanism more transparent.

4. The evaluation focuses on a single type of learnable additive-noise trigger inspired by BadCLIP. It does not test against adaptive or structurally different triggers (e.g., patch-based, frequency-domain, sample-specific, or model-poisoning attacks). As a result, the claimed generality of SABRE-FL across attack mechanisms is not sufficiently demonstrated.

**Questions:**

Please revise the methodology section based on the weaknesses mentioned above. Also, adding some more important experiments should strengthen the contribution of this paper.

---

> ### Author Response · Authors · 2025-11-20
> **Rebuttal by Authors (Part 1/4)**
>
> We thank the reviewer for acknowledging the timeliness of our setting and the clear motivation. We address the concerns point-by-point below.
>
> ## Trigger optimization details
> We apologize for the earlier omission and clarify the details here. The trigger is implemented as a learnable additive noise pattern (denoted δ), jointly optimized with the prompt vector using CLIP’s contrastive loss, following Bai et al. (BadCLIP, CVPR 2024). We use SGD with a cosine annealing scheduler and optimize δ for 3 epochs during the local training of malicious clients. The objective maximizes similarity between the poisoned image embedding and a target class prompt, while preserving clean image alignment. Unless stated otherwise, we use ViT-B/16 as the image encoder backbone, a fixed learning rate of 0.1 during trigger warm-up, and set ε = 4 (scaled to [0, 1] range) as the maximum allowed noise magnitude.
>
> ## Choice of m and client removal sensitivity
> Our original design followed a standard assumption in many FL defenses (e.g., Trimmed Mean, Median, Krum) that the defender can estimate m, allowing direct and fair comparison under this common threat model.
> However, we agree that SABRE-FL’s performance may depend on accurate knowledge of the attacker budget m, and we thank you for highlighting this. In response, we have conducted new ablation studies (Tables 7 and 8) to evaluate SABRE-FL under both underestimation and overestimation scenarios, where some malicious clients are left unfiltered or a few benign clients are mistakenly removed. Results show that even when some attackers remain in the aggregation, the defense still suppresses backdoor accuracy meaningfully; similarly, over-pruning a small number of benign clients does not significantly degrade clean performance. These findings suggest SABRE-FL is resilient to practical misestimations of m.
>
> **Table: Effect of imperfect client filtering (Table 7, Appendix F)**
> We intentionally do *not* remove some malicious clients and evaluate SABRE-FL’s robustness on **Pets** and **DTD**. Clean accuracy (CA) drops mildly, while backdoor accuracy (BA) rises with more undetected attackers.
>
> | #  Malicious Clients Not Removed |   |   |   | Pets |      |   |   |   | DTD  |      |
> |----------------------------------|---|---|---|------|------|---|---|---|------|------|
> |                                  |   |   |   | CA   | BA   |   |   |   | CA   | BA   |
> |                 1                |   |   |   | 92.1 | 4.7  |   |   |   | 62.2 | 11.3 |
> |                 2                |   |   |   | 91.1 | 7.2  |   |   |   | 61.0 | 14.1 |
> |                 3                |   |   |   | 90.5 | 9.1  |   |   |   | 61.1 | 19.7 |
> |                 4                |   |   |   | 89.8 | 10.1 |   |   |   | 60.5 | 25.5 |
>
> **Table: Overestimation, i.e., removing benign clients too (Table 8, Appendix F)**
> | Setting                        |   |   |   | Caltech |     |   |   |   | Pets |     |   |   |   | DTD  |     |   |   |   | Aircraft |      |   |   |   | Food101 |     |
> |--------------------------------|---|---|---|---------|-----|---|---|---|------|-----|---|---|---|------|-----|---|---|---|----------|------|---|---|---|---------|-----|
> |                                |   |   |   | CA      | BA  |   |   |   | CA   | BA  |   |   |   | CA   | BA  |   |   |   | CA       | BA   |   |   |   | CA      | BA  |
> | 2 Malicious + 1 Benign Removed |   |   |   | 97.3    | 6.6 |   |   |   | 94.8 | 4.5 |   |   |   | 64.5 | 6.8 |   |   |   | 33.4     | 0.4  |   |   |   | 90.5    | 2.0 |
> | 2 Malicious + 2 Benign Removed |   |   |   | 97.4    | 7.3 |   |   |   | 94.9 | 4.8 |   |   |   | 63.5 | 5.4 |   |   |   | 31.1     | 17.2 |   |   |   | 90.3    | 2.1 |

---

> ### Author Response · Authors · 2025-11-20
> **Rebuttal by Authors (Part 2/4)**
>
> ## Auxiliary dataset justification
> Thank you for this thoughtful observation. The goal of our detector is not to memorize dataset-specific patterns, but to capture a statistical signal induced by the backdoor trigger in the CLIP embedding space. As shown in our t-SNE visualizations (Figures 4 and 7), backdoored embeddings exhibit a consistent shift across datasets, even though the detector is trained on another dataset. This supports our key assumption: if a trigger is strong enough to fool a downstream classifier, it must consistently alter the image embedding to resemble that of the target class. SABRE-FL simply learns to recognize this shift. While we acknowledge that full generalization to all trigger types cannot be guaranteed, our empirical results suggest that the learned detector captures trigger-induced embedding deviations in a task-agnostic manner. We will emphasize this more clearly in the revision. However, to strengthen our claim, we select the Flowers dataset for detector training and re-run our experiments for detection. We show these results here and observe that the detector does indeed generalize across datasets because it learns to detect backdoor noise that is not specific to any dataset or class.
>
> ## Auxiliary dataset generalization
> Thank you for highlighting this important concern. To address this, we conducted new experiments where the detector is trained on a completely different dataset (Flowers‑102) and evaluated on the remaining benchmarks. As shown in Table 9, SABRE‑FL continues to achieve strong clean accuracy and suppress backdoor success, despite the change in label space and distribution. These results suggest that the detector does not memorize dataset-specific patterns but instead learns a transferable statistical signature of poisoned embeddings, supporting our claim of dataset-agnostic generalization.
>
> **Table: Evaluating SABRE-FL when the detector is trained on Flowers instead of Caltech. Despite the shift in auxiliary dataset, the defense maintains strong generalization across tasks, reducing backdoor accuracy (BA) while preserving clean accuracy (CA). (Table 9 in Appendix F)**
> | Setting                       |   |   |   | Caltech |         |   |   |   | Pets |         |   |   |   | DTD  |         |   |   |   | Aircraft |         |   |   |   | Food101 |         |
> |-------------------------------|---|---|---|---------|---------|---|---|---|------|---------|---|---|---|------|---------|---|---|---|----------|---------|---|---|---|---------|---------|
> |                               |   |   |   | CA      | BA      |   |   |   | CA   | BA      |   |   |   | CA   | BA      |   |   |   | CA       | BA      |   |   |   | CA      | BA      |
> | No Defense                    |   |   |   | 97.2    | 58.2    |   |   |   | 92.1 | 12.1    |   |   |   | 67.6 | 27.3    |   |   |   | 32.5     | 91.5    |   |   |   | 90.0    | 20.6    |
> | SABRE-FL (Trained on Flowers) |   |   |   | 97.1    | **6.1** |   |   |   | 94.6 | **4.4** |   |   |   | 64.4 | **4.7** |   |   |   | 32.0     | **0.2** |   |   |   | 90.7    | **2.2** |
>
> ## Scope of Evaluated Attack Types
> We sincerely appreciate the reviewer’s suggestion regarding broader attack evaluations. We agree that testing SABRE-FL against structurally diverse backdoor types (e.g., patch-based, frequency-domain, model-poisoning) would provide additional insight into its generality. However, we intentionally chose to fix the attack type, using a well-motivated, learnable additive trigger inspired by BadCLIP, to better isolate the impact of other variables such as defense strategies, client ratios, auxiliary datasets, and prompt configurations. Varying both attack and defense dimensions simultaneously would make it difficult to draw controlled insights or clearly attribute performance shifts. We view this paper as a focused first step on prompt-specific backdoor vulnerabilities in FL, and hope to explore other attack surfaces, including model poisoning, in follow-up work. We will explicitly clarify this scope and limitation in the final version.

---

> ### Author Response · Authors · 2025-11-20
> **Rebuttal by Authors (Part 3/4)**
>
> ## Evaluation under Non-IID settings
> We thank the reviewer for emphasizing the importance of evaluating defenses under realistic data heterogeneity. In response, we conducted a comprehensive set of experiments using Dirichlet-based partitioning with varying concentration parameters (α ∈ {0.9, 0.7, 0.5, 0.3, 0.1}) to simulate increasing levels of non-IID-ness. As shown in Table 4, SABRE-FL consistently reduces backdoor accuracy (BA) across all heterogeneity levels while preserving high clean accuracy (CA). Notably, under extreme heterogeneity (α = 0.1), there is a slight drop in defense performance, particularly for complex datasets like Food101 and Aircraft, but the defense still significantly outperforms the no-defense baseline. These results confirm that SABRE-FL remains robust under moderate to high heterogeneity, though future work may explore refinements to handle extreme distribution shifts even more effectively.
>
> **Table: Effect of varying Dirichlet Distribution, i.e., non-IID-ness (Table 4, Appendix F)**
>
> **_Due to space-constraints, Caltech is written as Calt. Aircraft is written as Airc. No-Defense is ND. Sabre-FL is SF_.**
> | $\alpha$ |   |   |   | **Calt** |      |   |      |      |   |   |   | **Pets** |      |   |      |      |   |   |   | **DTD** |      |   |      |      |   |   |   | **Airc** |      |   |      |      |   |   |   | **Food** |      |      |      |
> |----------|---|---|---|----------|------|---|------|------|---|---|---|----------|------|---|------|------|---|---|---|---------|------|---|------|------|---|---|---|----------|------|---|------|------|---|---|---|----------|------|------|------|
> |          |   |   |   | ND       |      |   | SF   |      |   |   |   | ND       |      |   | SF   |      |   |   |   | ND      |      |   | SF   |      |   |   |   | ND       |      |   | SF   |      |   |   |   | ND       |      | SF   |      |
> |          |   |   |   | CA       | BA   |   | CA   | BA   |   |   |   | CA       | BA   |   | CA   | BA   |   |   |   | CA      | BA   |   | CA   | BA   |   |   |   | CA       | BA   |   | CA   | BA   |   |   |   | CA       | BA   | CA   | BA   |
> | 0.9      |   |   |   | 97.7     | 22.3 |   | 97.7 | 8.4  |   |   |   | 95.2     | 5.8  |   | 95.3 | 4.6  |   |   |   | 64.9    | 9.3  |   | 64.5 | 7.1  |   |   |   | 31.9     | 86.7 |   | 32.5 | 0.1  |   |   |   | 90.0     | 36.0 | 90.4 | 3.0  |
> | 0.7      |   |   |   | 97.3     | 22.1 |   | 97.2 | 8.5  |   |   |   | 94.6     | 10.8 |   | 94.4 | 4.9  |   |   |   | 61.8    | 11.3 |   | 64.1 | 9.6  |   |   |   | 30.7     | 85.0 |   | 30.9 | 33.0 |   |   |   | 90.0     | 14.7 | 90.3 | 2.4  |
> | 0.5      |   |   |   | 97.2     | 30.9 |   | 97.7 | 8.3  |   |   |   | 94.4     | 17.9 |   | 95.3 | 4.7  |   |   |   | 62.2    | 25.3 |   | 65.0 | 7.6  |   |   |   | 30.7     | 85.7 |   | 30.0 | 4.4  |   |   |   | 89.8     | 36.1 | 89.5 | 33.2 |
> | 0.3      |   |   |   | 97.5     | 32.6 |   | 97.4 | 20.5 |   |   |   | 93.0     | 12.2 |   | 91.0 | 10.0 |   |   |   | 63.5    | 9.6  |   | 60.2 | 11.0 |   |   |   | 30.3     | 86.7 |   | 31.5 | 83.9 |   |   |   | 89.7     | 71.6 | 89.2 | 19.9 |
> | 0.1      |   |   |   | 97.0     | 24.5 |   | 96.7 | 13.8 |   |   |   | 94.3     | 12.8 |   | 92.8 | 7.3  |   |   |   | 60.6    | 12.6 |   | 59.0 | 11.1 |   |   |   | 31.7     | 81.3 |   | 30.9 | 89.6 |   |   |   | 89.4     | 56.7 | 89.4 | 37.3 |

---

> ### Author Response · Authors · 2025-11-20
> **Rebuttal by Authors (Part 4/4)**
>
> ## Trigger optimization steps
> We thank the reviewer for this important suggestion. In response, we conducted two sets of ablations to assess how trigger tuning affects attack efficacy and defense performance.
> First, we varied the number of trigger optimization steps (1, 3, and 5 epochs) to study how training duration influences backdoor strength. We observed that increasing optimization epochs consistently enhances the attack's potency, as reflected in higher backdoor accuracy when no defense is applied. This confirms that the optimization step is a key factor in trigger effectiveness.
>
> **Table: Effect of trigger optimization steps, i.e. epochs (Table 6, Appendix F)**
> | Setting    |   |   |   | Trigger optimization |   |   |   | Caltech |      |   |   |   | Pets |     |   |   |   | DTD  |      |   |   |   | Aircraft |      |   |   |   | Food |      |
> |------------|---|---|---|----------------------|---|---|---|:-------:|:----:|---|---|---|------|-----|---|---|---|------|------|---|---|---|----------|------|---|---|---|------|------|
> |            |   |   |   |                      |   |   |   | CA      | BA   |   |   |   | CA   | BA  |   |   |   | CA   | BA   |   |   |   | CA       | BA   |   |   |   | CA   | BA   |
> | no defense |   |   |   | 1 epoch              |   |   |   | 96.9    | 31.6 |   |   |   | 94.1 | 7.0 |   |   |   | 64.7 | 30.3 |   |   |   | 28.9     | 61.7 |   |   |   | 90.2 | 3.0  |
> | no defense |   |   |   | 5 epochs             |   |   |   | 97.2    | 58.4 |   |   |   | 94.6 | 6.3 |   |   |   | 61.9 | 41.8 |   |   |   | 31.3     | 87.6 |   |   |   | 90.1 | 22.6 |
> | SABRE-FL   |   |   |   | 1 epoch              |   |   |   | 97.1    | 7.8  |   |   |   | 94.4 | 4.2 |   |   |   | 67.8 | 6.2  |   |   |   | 31.6     | 3.2  |   |   |   | 90.4 | 2.2  |
> | SABRE-FL   |   |   |   | 5 epochs             |   |   |   | 97.3    | 8.1  |   |   |   | 94.7 | 4.3 |   |   |   | 66.2 | 2.7  |   |   |   | 31.0     | 0.0  |   |   |   | 90.5 | 3.1  |
>
> ## Trigger Strength
> Second, we evaluated the impact of trigger strength by scaling the ε parameter, which controls the pixel-level perturbation budget. We tested with ε/2 (weaker) and 2×ε (stronger), and observed that stronger triggers lead to more successful attacks. Importantly, however, SABRE-FL remains robust under both settings, maintaining low backdoor accuracy even against amplified triggers. Detailed results are included in Table 5 and Table 6.
>
> **Table: Effect of trigger strength, i.e., ϵ scaling (Table 5, Appendix F)**
> | Setting    |   |   |   | Trigger Strength |   |   |   | Caltech |         |   |   |   | Pets |         |   |   |   | DTD  |         |   |   |   | Aircraft |         |   |   |   | Food101 |         |
> |------------|---|---|---|------------------|---|---|---|---------|---------|---|---|---|------|---------|---|---|---|------|---------|---|---|---|----------|---------|---|---|---|---------|---------|
> |            |   |   |   |                  |   |   |   | CA      | BA      |   |   |   | CA   | BA      |   |   |   | CA   | BA      |   |   |   | CA       | BA      |   |   |   | CA      | BA      |
> | No Defense |   |   |   | 2x epsilon       |   |   |   | 97.2    | 51.4    |   |   |   | 93.9 | 6.0     |   |   |   | 64.9 | 51.4    |   |   |   | 31.2     | 92.7    |   |   |   | 90.1    | 33.5    |
> | No Defense |   |   |   | 0.5x epsilon     |   |   |   | 97.1    | 8.7     |   |   |   | 92.1 | 27.3    |   |   |   | 64.6 | 27.8    |   |   |   | 30.5     | 80.0    |   |   |   | 89.8    | 4.5     |
> | SABRE-FL   |   |   |   | 2x epsilon       |   |   |   | 97.1    | 7.6     |   |   |   | 94.6 | 4.6     |   |   |   | 64.4 | 4.6     |   |   |   | 32.0     | 17.2    |   |   |   | 90.7    | 3.9     |
> | SABRE-FL   |   |   |   | 0.5x epsilon     |   |   |   | 97.1    | 8.3 |   |   |   | 94.6 | 4.3 |   |   |   | 64.4 | 5.8 |   |   |   | 32.0     | 1.0 |   |   |   | 90.7    | 2.1 |
>
> ## Trigger Visualization
> We also appreciate the suggestion to visualize the learned triggers. We have now included representative samples from all datasets, showcasing clean, backdoored, and trigger-only images (see **Figure 9 in Appendix F**). These examples demonstrate that while the perturbations remain visually subtle, they are sufficient to induce misclassification, supporting our claim that the attack is visually imperceptible but semantically effective.

---

### Author Response · Authors · 2025-11-20
**Global Rebuttal Summary by Authors**

We sincerely thank all reviewers for their thoughtful feedback and constructive comments. We are encouraged by the recognition of the paper’s novelty and relevance, and we have addressed each reviewer’s concerns in detail. Several reviewers highlighted important aspects regarding evaluation breadth, practical deployment, and robustness to adaptive settings. We have carefully considered these suggestions and conducted a number of new experiments to clarify our contributions and limitations.

Below, we summarize the key concerns raised and our responses:

* **Non-IID Data and Heterogeneity Effects:** A recurring concern was SABRE-FL’s robustness under federated data heterogeneity. We now include a comprehensive Dirichlet-based study (Appendix F, Table 4), varying $\alpha$ from 0.9 (mild) to 0.1 (severe). Our results show SABRE-FL remains effective across most settings, with minor performance drops only at extreme heterogeneity levels.

* **Sensitivity to m (Number of Malicious Clients):** As raised by multiple reviewers, we evaluated SABRE-FL under both underestimation (failing to remove attackers) and overestimation (removing benign clients). The defense is fairly robust under both scenarios (Appendix F, Tables 7 & 8).

* **Qualitative Visualization:** We added visual examples of clean, backdoored, and triggered images across all datasets (Appendix F, Figure 9), showing the imperceptibility of triggers and supporting the paper’s claims.

* **Trigger Strength and Optimization:** We conducted ablations varying both the trigger magnitude ($\epsilon$) and the optimization duration. These show a clear trend: stronger triggers result in higher attack success, but SABRE-FL consistently mitigates the backdoor (Appendix F, Tables 5 & 6).

* **Generalization Across Datasets:** To verify that SABRE-FL’s embedding-space detector does not overfit to a particular auxiliary dataset, we trained the classifier on a different OOD dataset (Flowers-102 instead of Caltech-101) and re-ran our main experiments. The detector generalized well across five target datasets, reaffirming that it captures dataset-agnostic trigger signatures (Appendix F, Table 9).

* **Practicality of the Detector Pretraining Phase:** Multiple reviewers questioned how the server obtains poisoned examples for training the anomaly detector. In response, we clarified that poisoned embeddings can be synthetically generated offline using public OOD data and a generic trigger. This setting mirrors realistic FL defenses where the server simulates threat scenarios during calibration.

* **Comparison with FLAME and Related Work:** We clarified conceptual differences with FLAME and related defenses. SABRE-FL leverages a supervised embedding-space detector, offering precise client-level filtering, while being tailored to prompt-based FL systems.

* **Adaptive Attackers:** While full adaptive attack analysis is left to future work, we note that the robustness under misestimated $m$ indirectly simulates such scenarios, where a smart attacker evades detection. Our results suggest SABRE-FL still limits backdoor success in such cases.

We believe the additional results and clarifications significantly strengthen the submission. All new results are included in Appendix F. We will integrate key insights and references in the revised paper and will be grateful for the opportunity to update the paper for acceptance at ICLR'26. Thank you again for your detailed reviews.

---

### Meta-Review · Area_Chair_UXch · 2026-01-06

**Summary:**

The paper studies backdoor attacks and defenses in the context of federated prompt learning. The authors construct a backdoor attack that works in FL settings where an adversary controls a fraction of the clients. They also propose a defense that detects shifts in embedding space and measures its effectiveness on several datasets and against several robust aggregation methods.

Overall, the reviewers praised the novel topic and appreciated the simplicity of the presented attack and defense, which also seem to be the first backdoor methods in such contexts.

Still, the reviewers highlighted several important details and experiments that were missing  (e.g. details about the attack (Reviewer D5Ed), as well as the claimed indistinguishability of the backdoored examples and experiments with non-IID data (several reviewers)). The authors did provide additional information and ablations, which should be included in the next version of the manuscript. Additionally, the defense method remains specific to the proposed attack (the auxiliary dataset is constructed via inserting the triggers from the proposed attack) and its hard to see if the defense will generalize to other attacks or even variants of the proposed attack.

**Reviewer Concerns:**

The reviewers asked for quite a few ablations and the authors did provide many of these. Two reviewers opted to keep their score of weak accept.

**Reviewer Scores:**

Two reviewers with weak accept opted to keep their score. Another one with the same score has similar concerns, so it is unlikely they would have raised. Finally, reviewer D5Ed did not respond before the closing of discussions. The authors did provide further clarifications, several of these are central to the core paper contribution and therefore should be included in the next version of the paper.

---

### Decision · Program_Chairs · 2026-01-26

Accept (Poster)